# LLM-BASED AUTOMATED THEOREM PROVING HINGES ON SCALABLE SYNTHETIC DATA GENERATION

## ABSTRACT

Recent advancements in large language models (LLMs) have sparked considerable interest in automated theorem proving and a prominent line of research integrates stepwise LLM-based provers into tree search. In this paper, we introduce a new data synthesis method that explores a wide range of intermediate proof states to generate diverse proof steps, which facilitates effective one-shot fine-tuning of the LLM policy model. We also propose an adaptive beam size strategy, which effectively takes advantage of our data synthesis method and achieves a trade-off between exploration and exploitation during tree search. Evaluations on the MiniF2F and ProofNet benchmarks demonstrate that our method outperforms strong baselines under the stringent *Pass@1* metric, attaining an average pass rate of $60.74\%$ on MiniF2F and $21.18\%$ on ProofNet. These results underscore the impact of large-scale synthetic data in advancing automated theorem proving.

## 1 INTRODUCTION

Reasoning has emerged as a frontier in large language model (LLM) research. Numerous approaches have been proposed to enhance their reasoning capabilities (Yu et al., 2024; Giadikiaroglou et al., 2024), among which mathematical reasoning (Ahn et al., 2024) has received special attention from both academia and industry. The reasons are twofold: (1) it is widely considered as a strong indicator of LLM's cognitive proficiency; and (2) it has a direct application in the emerging AI4Math area and great potential in software engineering (Li et al., 2024; Liu et al., 2024). While extensive research has focused on solving mathematical problems given in natural language (Muennighoff et al., 2025; Xiang et al., 2025), we are mostly interested in proving theorems formulated in formal mathematical languages, because it represents a more fundamental challenge. To this end, we synergize interactive theorem provers (ITPs, aka proof assistant), represented by Lean (Moura & Ullrich, 2021), Isabelle (Paulson, 1994) and Rocq (previously known as Coq (Barras et al., 1997)), and LLMs, effectively giving rise to a neuro-symbolic approach towards automated theorem proving (ATP).

In the literature, there are generally two classes of approaches which harness LLMs for ATP, commonly referred to as *tree search* methods and *whole-proof generation* methods (Ren et al., 2025). The former repeatedly performs proof step generation, utilizing LLMs to prescribe a tactic to be applied to the current proof state. Overall, it is cast as a tree search process to generate the final proof. In contrast, the latter is performed in an end-to-end style, where LLMs are asked to produce an entire proof directly from the theorem, which is then verified by the proof assistant. Generally speaking, the whole-proof methods are simpler, requiring less communication to coordinate between the LLM and the ITP. However, as the intermediate proof states are hidden, LLMs typically struggle to generate a long proof. In practice, these methods usually rely heavily on additional techniques, such as natural language comments to assist the proof (Lin et al., 2025; Xin et al., 2024b).

The class of tree search approaches has its advantages in demanding relatively lower capability from the policy model, making it suitable for settings with smaller model sizes and limited computational resources. It also offers greater flexibility for incorporating customized, problem-specific strategies into the search process (Wang et al., 2024; An et al., 2024). Furthermore, starting from a small number of seed problems, tree search enables extensive exploration of proof states, facilitating the synthesis of large volumes of new data for iterative training the policy model, which is considerably challenging to achieve in whole-proof generation (Xin et al., 2024a). We take Lean 4 as the base

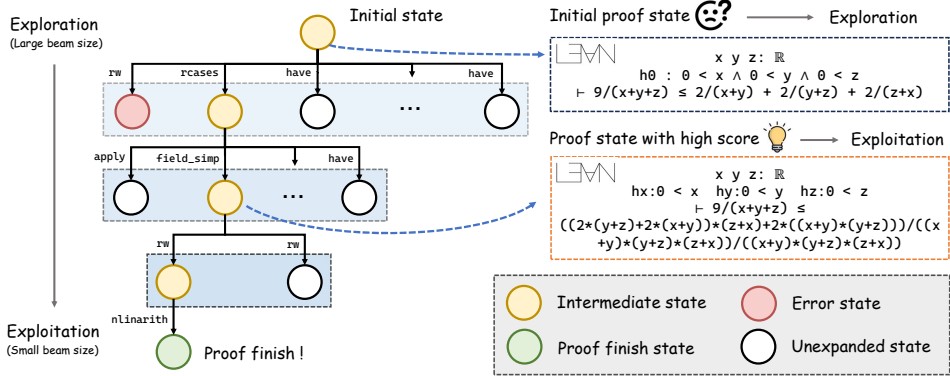

Figure 1: An illustration of the proposed tree search strategy. The example is derived from the MiniF2F benchmark, specifically the problem *algebra_9onxpypzleqsum2onxpy*. For simplicity, only the tactics are retained in the depicted proof states, while the associated premises are omitted. The basic functionality of the tactics used in our examples (e.g. `rw`, `rcases`, `have`, `apply`, and etc.) is further explained in Appendix A.9.

ITP due to its popularity in the community, although our methodology can be easily applied to other proof assistants.

However, LLMs still remain notably limited in formal mathematical reasoning (Yang et al., 2024b). A fundamental cause is that LLMs are typically trained for general-purpose language understanding rather than for theorem proving, which requires substantial domain expertise in mathematics. To overcome this gap, post-training is indispensable, for which we focus on fine-tuning. A major challenge lies in the scarcity of high-quality training data for theorem proving. Proofs in ITPs are typically expressed as code in a domain-specific language (DSL) defined by the corresponding prover. Compared to widely used programming languages such as Python, proofs written in Lean are extremely limited in volume. For instance, standard Lean 4 libraries (e.g., Mathlib (mathlib Community, 2020)) provide less than 1GB data in total, insufficient for effectively fine-tuning LLMs. As a result, synthetic data generation has become a crucial component in adapting LLMs for ATP. Recent efforts have explored translating natural language mathematics into Lean 4 syntax using LLMs (Wu et al., 2022; Ying et al., 2024; Murphy et al., 2024), generating new problems by sampling from distributions (e.g., a fine-tuned policy model) learned over existing Lean 4 formalizations (Dong & Ma, 2025; Poesia et al., 2024; Xin et al., 2024a). Another approach to improving model capability is Expert Iteration (EI) (Polu & Sutskever, 2020), which can be treated as a form of rejection sampling. Given a set of formalized mathematical problems, a model may fail to solve the more difficult instances but succeed on simpler ones. By collecting successful proofs from these solvable cases, a new dataset can be constructed to fine-tune the model, thereby gradually enhancing its capabilities. This method has been widely used in prior work (Polu & Sutskever, 2020; Polu et al., 2022; Wu et al., 2024; Xin et al., 2025). However, EI suffers from significant inefficiency when applied in conjunction with tree search. In each iteration, the number of newly solvable problems is often limited, while performing a full search over the entire dataset incurs substantial computational cost. This inefficiency makes EI impractical in settings with limited computational resources, highlighting *the need for a more direct and resource-efficient data generation strategy*.

In this paper, we propose a data synthesis method called *proof state exploration* for fine-tuning LLMs in ATP, along with a complementary *adaptive beam size strategy* for tree search during the proving process. Our data synthesis method begins with a set of formalized mathematical problems as seeds and leverages the existing policy model to explore related intermediate proof states via tree search. These intermediate states may correspond to goals that are logically equivalent to the original ones after a sequence of transformations, or to subgoals resulting from original problem decomposition. During the process, we explicitly decouple the generation of tactics from that of premises. Specifically, we adopt a constrained decoding algorithm (Hokamp & Liu, 2017) that restricts the model to sample from and de-duplicate within a curated set of commonly used tactics, followed by premise completion handled by the policy model. In addition, we introduce a heuristic

pruning strategy to balance the trade-off between diversity in the generated outputs and computational efficiency during the exploration. This enables large-scale synthesis of intermediate proof states and transformation steps within tree search, thereby reducing the distribution gap between the training data and real-world proving scenarios. Moreover, our method does not rely on EI. Instead, our policy model is trained on data generated from a single, exhaustive exploration over existing formalized problems. This one-pass generation approach significantly reduces computational overhead while achieving competitive performance.

Although the LLM fine-tuned on this synthetic dataset can serve as a policy model capable of generating diverse tactics, it may not always prioritize proof completion during actual theorem proving. To mitigate this limitation, we propose a simple yet effective *adaptive beam size strategy* tailored to our data synthesis framework and integrated into the tree search process (cf. Figure 1). The strategy starts with a relatively large beam size to encourage broad exploration of the space of proof states. As the search proceeds, the beam size is gradually reduced, enabling the policy model to concentrate on high-scoring proof states. This dynamic adjustment improves the focus of the search and enhances the overall success rate of proof completion. Besides, we develop Dojo-BeamSearch-Visualization (DoBeVi) (cf. Appendix A.4) as the basic interaction tool between the model and the Lean 4 prover with a visualization module for tree search analysis.

*Evaluation.* We evaluate our method on two widely adopted benchmarks, MiniF2F and ProofNet, achieving average pass rates of $60.74\%$ and $21.18\%$, respectively, under the constraints of *Pass@1* and a limited computational budget. These results outperform the current state-of-the-art methods based on tree search.

The main contributions of this paper are as follows.

- We propose a novel data synthesis method, *proof state exploration*, which enables large-scale generation of exploration data for fine-tune LLMs as policy models in ATP.

- We propose a simple but efficient *adaptive beam size strategy* that synergistically operates with the data synthesis method, effectively guiding the search process towards proof completion.

- We carry out extensive experiments demonstrating that our method can effectively balance exploration and exploitation during tree search, achieving state-of-the-art performance among existing tree search methods.

## 2 BACKGROUND

*ITP and ATP.* In a nutshell, interactive theorem proving is a process to develop and verify formal proofs with the assistance of a computer, usually within a proof assistant (e.g., Lean, Isabelle, Rocq), and involves human guidance to structure and complete the proof. The general process is to formalize the theorem to be proven (i.e., the goal) in the language of the proof assistant first, and then repeatedly apply tactics to break the goal into subgoals. The tactic may involve case analysis, induction, rewriting, application of lemmas or axioms, etc. As such, the human user iteratively refines the proof to guide the prover, choosing which rules to apply and in what order. The prover checks that each inference step is valid according to the rules of the formal system. A major drawback is that this process requires heavy human expertise and thus largely lacks automation. LLM-based Automated Theorem Proving aims to find a proof automatically by simulating this process, effectively replacing the human user by an LLM.

*Tree Search LLM-based ATP.* As mentioned in the Section 1, we adopt a tree search approach towards LLM-based ATP. Generally speaking, tree search is a class of decision-making algorithms which consists of searching combinatorial spaces represented by trees, where nodes denote states and edges denote transitions (actions) from one state to another. In our setting, they enable an efficient navigation of the large and complex proof space.

At a high level, tree search LLM-based ATP can be cast as a reinforcement learning (RL) problem, where the state space comprises the proof states (aka proof terms). In general, proof states represent the current status of the proof at any given point in the interactive session between LLM and the proof assistant. Typically, they comprise the context (i.e., a list of assumptions, hypotheses and local variables), the current (sub)goals (i.e., the proposition needing to prove to complete the proof) and the number of subgoals to be proven. LLM acts as an actor, which, given the current proof state $s_i$,

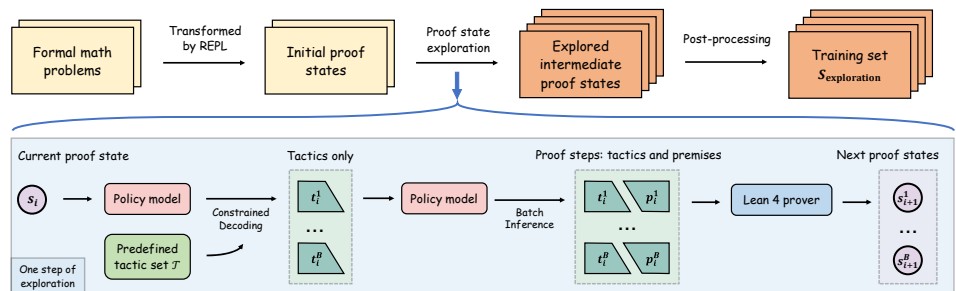

Figure 2: An illustration of the data synthesis pipeline.

generates a tactic. In other words, LLM serves as the *policy model*. In theorem proving, tactics refer to commands, or instructions, that guide the process of constructing a proof. In RL terms, tactics can be regarded as an action $a_i$, based on which the proof assistant produces a new proof state $s_{i+1}$. This process continues until either the theorem is successfully proved or the maximum number of iterations is reached.

There are a plethora of search strategies, for instance, Monte Carlo tree search (MCTS) or best-first search (BFS). Although MCTS is well suited for balancing exploration and exploitation, its classical simulation step is impractical in the context of ATP. Once a simulation reaches a successful proof state, the proof is already complete, and no further search is required. In BFS, each expansion step involves using the policy model to perform beam search decoding, generating a fixed number of top candidates (called the beam size). Typically, the process begins at the root node and iteratively expands the current node to produce potential child nodes. These candidates are then evaluated using a heuristic scoring function, and the node with the highest score is selected for the next expansion. The expansion-score-selection steps are repeated until a goal is reached or a termination condition is met. In our work, LLM, as the policy model, generates a set of tactics $\{t_i^j\}_{j=1}^B$ for a proof state $s_i$, which are executed by the proof assistant one by one to obtain the corresponding set of next proof states $\{s_{i+1}^j\}_{j=1}^B$.

## 3 METHODS

### 3.1 DATA SYNTHESIS BY PROOF STATE EXPLORATION

In ITP, tactics play a central role. Normally, a tactic (e.g., `rw [add_assoc add_comm]`) consists of the tactic operator (e.g., `rw`) and the premises (e.g., `add_assoc` and `add_comm`). In this paper, to facilitate a finer-grained decomposition, what is commonly referred to as tactics are called *proof steps*, while *tactic* refers to the operator only. We propose to carry out *proof state exploration* to synthesize data. This is similar to tree search, but the crucial difference is that we prioritize the diversity of tactics. Our data synthesis method is described as Algorithm 1. (cf. Figure 2 as well).

**Motivated explanation**. The intuition is that proof assistants (e.g., Lean 4) normally provide a versatile range of tactics, which is considerably larger than, e.g., the set of reserved words in programming languages, and can even be defined by users. It is practically infeasible to adopt all of them in tree search. On the other hand, in practice, their actual use in theorem proving may not be as wild as, e.g., user-defined functions (which are virtually infinite, or at least cannot be enumerated exhaustively). Hence, as a trade-off, a reasonably curated set of commonly used tactics is sufficient to solve the vast majority of problems in practice. Besides that, for tree search, critically we need an "effective" set of (proof state, tactic) pairs. The observation is that in theorem proving, the same tactic may have multiple meanings and can be used across various scenarios; for instance, the `apply` tactic may be used in five different scenarios.[1] Moreover, sometimes a tactic can be used in much broader cases; for instance, the tactic `rfl` can handle all *definitional equalities*. However, as the policy model, the LLM, like an inexperienced problem solver, may fail to realize that a tactic can actually be used

---

[1]`https://www.ma.imperial.ac.uk/~buzzard/xena/formalising-mathematics-2024/Part_C/tactics/apply.html#`

when a proof state is encountered. Ideally, such domain-specific knowledge should be injected into LLMs via training, as they largely fall short of a certain level of abstraction capabilities and so would be hard to learn via the standard tree search.

**The proposed proof state exploration method**. We intend to identify those tactics which occur rarely (with low probability) at a proof state. Just like a human problem solver normally has the "comfort zone" for certain methods when facing a problem, in tree search, the policy model likely picks up the tactic which has a higher probability. Quite often, this is sub-optimal or may represent a missed opportunity. Our proof state exploration may force the policy model to explore those tactics which it does not typically explore (i.e., force it to jump out of its comfort zone), by which a more diverse training set can be constructed.

To this end, we first identify the frequently used tactics. We scan the entire Mathlib to collect all defined tactics.[2] There are hundreds of them, most of which are rarely used. To filter these out, we need a representative dataset from which the frequency of each tactic can be collected. In our work, we use the STP dataset (Dong & Ma, 2025) from which a total of 173 distinct tactics are obtained. To further refine the set, we apply nucleus sampling with a threshold of $\mathcal{P} = 0.999$ to filter out rarely used tactics, ultimately yielding a set $\mathcal{T}$ of 60 commonly used tactics.

The design of the proof state exploration is guided by the above intuition and resembles the tree search. For each expansion step, we apply Constrained Decoding to force the LLM to output $B$ different tactics (without premises) in $\mathcal{T}$. We set a large beam size (e.g., $B = 32$). After obtaining $\{(s_i, t_i^j)\}_{j=1}^B$ with $t_i^j \in \mathcal{T}$, we feed these pairs back into the policy model to complete the premises, yielding a set $\{(s_i, t_i^j, p_i^j)\}_{j=1}^B$ of proof steps from $s_i$. By applying each $(t_i^j, p_i^j)$ to $s_i$ using Lean 4 prover, we generate the corresponding set of next proof states $s_{i+1}^j$. Note that for exploration, even if *proof finish* state is encountered, the exploration continues until the entire budget is exhausted.

Potential issues for this proof state exploration include (1) as $B$ is often large, obtaining deeper proof states (i.e., those "far away" from $s_0$) can take a long time under limited computational resources; (2) in certain proof states, a large beam size can lead to the generation of meaningless proof steps. For example, the model may construct hypotheses that do not contribute to the progress of the proof.

To address these, we apply two heuristic strategies: (1) During each expansion, we prune the set by retaining only $\alpha B$ branches, typically setting $\alpha = 0.25$. When pruning, we keep the top $\beta$ branches with the highest joint probabilities and then randomly sample the remaining $(\alpha B - \beta)$ branches uniformly from the rest. This balances exploration diversity and efficiency while preserving some of the policy model's inherent tendency toward proof finish. (2) Some seeds are overly simple and do not require extensive exploration; too much exploration may even degrade the quality of the synthetic data. To control this, we use proof finish as a signal: if a distinct path achieving proof finish is found during an expansion, we reduce the expansion budget by a factor of $\gamma = 0.9$. This allows a simple seed to quickly terminate exploration after generating multiple distinct solutions, improving overall data synthesis efficiency.

After the proof state exploration, we obtain a dataset $S_{\text{exploration}}$. To produce the final dataset, we apply three post-processing steps: (1) De-duplication, to remove redundant samples; (2) Decontamination, where we use BLEU as a similarity metric to eliminate examples with high overlap with the evaluation benchmark; and (3) Rejection Sampling, where we discard all transitions that are invalid from the Lean 4 prover. The generated data is ultimately used to train policy LLM by performing Supervised Fine-Tuning on a base model using the full accumulated training data corpus.

### 3.2 ADAPTIVE BEAM SIZE STRATEGY

The tree search approaches used for ATP are conceptually similar (Polu & Sutskever, 2020; Polu et al., 2022; Wu et al., 2024; Xin et al., 2025), differing mainly in (1) the policy model, (2) the scoring function, (3) the beam size, and (4) other budget-related parameters. In our tree search procedure, the policy model is trained using data collected by the proof state exploration, as in Section 3.1. For the score function, we take the sum of the LLM's logarithmic joint probability over the beam (during beam search) and the parent node's score as the current node's score, that is:

$$\text{score}(s_{i+1}^j) = \log P_\theta(t_j|s_i) + \text{score}(s_i) \tag{1}$$

---

[2]https://github.com/haruhisa-enomoto/mathlib4-all-tactics

---

**Algorithm 1:** Proof State Exploration Algorithm for Data synthesis.

---

**Data:** $S = \{s_{0,n}\}_{n=1}^{|S|}$;     /\* $S$ is the original formalized datasets.   \*/
**Result:** $S_{\text{exploration}}$
$S_{\text{exploration}} \leftarrow \emptyset$;
**for** $s_{0,n} \in S$ **do**
    $s_0 \leftarrow s_{0,n}$;                    /\* ignore index $n$ for simplicity.   \*/
    $q \leftarrow \emptyset$;                              /\* $q$ is a heap.   \*/
    $b \leftarrow 0$ ;                    /\* $b$ is the consumed budget.   \*/
    $q$.add($s_0$);
    **while** $|q| > 0$ **and** $b < budget$ **do**
        $s_i \leftarrow q$.get();
        $\{t_i^j\}_{j=1}^B \leftarrow$ policy\_model.constrained\_decoding($s_i, \mathcal{T}$);
        $\{p_i^j\}_{j=1}^B \leftarrow$ policy\_model.batch\_inference($s_i, \{t_i^j\}_{j=1}^B$);
        **for** $j \in [1, B]$ **do**
            $s_{i+1} \leftarrow$ lean\_prover.run\_step($s_i, t_i^j, p_i^j$); /\* run proof step and get new state $s_{i+1}$ from Lean prover \*/
            $S_{\text{exploration}}$.add($(s_{i+1}, t_i^j, p_i^j, s_i)$);
            **if** $s_{i+1}$ *is a valid proof state* **then**
                $q$.add($s_{i+1}$);
            **end**
        **end**
        $b$.add\_budget();
    **end**
**end**
$S_{\text{exploration}} \leftarrow$ post\_processing($S_{\text{exploration}}$)

---

where $P(t_j|s_i)$ is the model's predicted probability of applying the tactic $t_j$ at the state $s_i$ and $\theta$ denotes the policy models parameters.

In this work, we focus the pruning strategy on dynamically decaying the *beam size* during the tree search process. This design is motivated by the empirical finding that beam size has a particularly significant impact on overall performance compared to other factors in tree search. The details of the finding are presented in Section 4.3. The optimal beam size is difficult to determine in advance. The general observation is that, if the beam size is too small, the search may fail to discover proof states, missing potential opportunities; if the beam size is too large, the search may risk being trapped within one sub-tree rooted at a high-scoring intermediate node, resulting in futile expansions. Hence, it is necessary to adapt the beam size to better control the tree search.

We adopt a larger beam size at the early stages of tree search; as the search proceeds (i.e., the depth increases), we gradually reduce the beam size. This is largely aligned with the aforementioned observation. In particular, during the later stages of tree search, being trapped is particularly harmful as it may dump the search budget. By reducing the beam size, we effectively prune away less relevant branches and prevent the search from over-committing to one sub-tree too much. One may intuitively understand the design by taking an analogy of an IMO contestant. At the early stage, (s)he may want to widen the horizon by exploring different problem-solving strategies (e.g., reading books, surfing AoPS). However, when sitting in the competition (i.e., at a later stage), (s)he would better focus on relatively small strategies to attempt, prioritizing solving the problems successfully. To this end, we design an adaptive beam size function, namely,

$$\text{beam\_size} = B_{\min} + (B_{\max} - B_{\min}) \times \max(1 - \lambda M, 0) \tag{2}$$

where $M = e/E$, with $e$ denoting the current expansion step and $E$ representing the maximum number of expansions allowed by the budget. The parameter $\lambda$ is a hyperparameter controlling the decay rate of the beam size. The beam size gradually decreases from $B_{\max}$ to $B_{\min}$ as the search proceeds, thereby realizing the adaptive behavior described earlier. This search dynamic has also been confirmed empirically (cf. Figure 3 in Appendix).

## 4 EXPERIMENTS

### 4.1 EVALUATION SETTINGS

Below are the basic experimental settings (cf. Section A.5 for more details). All experiments were conducted on a server with 8 H800 GPUs; the code and model checkpoint are available[3].

**Models.** We use Qwen2.5-Math-7B (Yang et al., 2024a) as the base model to train the policy model. During full fine-tuning, we use a batch size of 2,048 and a learning rate of $2 \times 10^{-5}$, training for one epoch. A cosine annealing schedule is employed for the learning rate, reaching $10\%$ of its initial value by the end of the epoch. Notably, the model training process does *not* incorporate EI.

**Datasets.** We use the STP dataset (Dong & Ma, 2025) as the seed and adopt BFS-Prover (Xin et al., 2025) as the policy model for *proof state exploration*. We first generate the synthetic dataset following the method in Section 3.1. Then, before the actual training begins, the dataset is further combined with human-authored data from Mathlib, followed by deduplication and decontamination. This results in a final training set comprising approximately 20 million proof transitions.

**Baselines.** We select two state-of-the-art models that also use tree search methods, InternLM2.5-StepProver (Wu et al., 2024) and BFS-Prover (Xin et al., 2025), as main baseline models, applying the same search budget in our experiments.

**Environments.** We develop DoBeVi (cf. Appendix A.4) as the basic interaction tool between the model and the Lean 4 prover (v4.10.0).

**Benchmarks, metrics and computation budgets.** We evaluate on two popular benchmarks, MiniF2F (Zheng et al., 2021) and ProofNet (Azerbayev et al., 2023a), using *Pass@1* as the metric. We follow the commonly adopted search budget in tree search methods: $K \times B \times E$, where $K$ is the number of searches per theorem (i.e., $k$ in *Pass@k*); $B$ is the beam size per expansion; and $E$ is the maximum number of expansions allowed in a single search. For all baselines, the beam size in the search budget follows the configurations specified in the original papers, which we consider to be a fair setting. Further discussions are provided in Section 4.3.

### 4.2 MAIN RESULTS

Table 1 presents the results of our model alongside baseline methods on MiniF2F and ProofNet. We fix $K = 1$ and $E = 600$ and adopt the beam sizes used in the original baseline papers. If a baseline has reported results under the same computational budget, we directly use those results; otherwise, we re-evaluate the baseline using our own tree search implementation under identical experimental settings, including additional search techniques detailed in Appendix A.5.

The results demonstrate that our method achieves better evaluation performance even under a relatively constrained computational budget without relying on EI. Moreover, under the fixed beam size setting, the optimal beam sizes for our policy model are found to be 8 on MiniF2F and 32 on ProofNet, respectively. By adopting our adaptive beam size strategy, we are able to further improve performance without searching for the best beam size settings.

### 4.3 DISCUSSIONS

Unlike whole-proof generation methods, the performance of tree search methods is influenced by a variety of factors, in addition to the policy model. We found that the beam size is particularly critical, sometimes even comparable in importance to the policy model's capability. As shown in Table 1, both our model and the baselines exhibit different optimal beam size choices across the MiniF2F and ProofNet benchmarks. We attribute this primarily to differences in training data distribution. Currently, most publicly available Lean 4 training datasets are derived from Lean-Workbook (Ying et al., 2024), whose distribution resembles the MATH subset of MiniF2F. Consequently, using a larger beam size in such settings tends to introduce redundant search branches, causing the search to waste the budget without improving results. In contrast, for benchmarks such as ProofNet, which contain more out-of-distribution samples, increasing the beam size is necessary, which allows the

---

[3]The implementation code is accessible at `https://github.com/RYbWnlL0v8/llm_atp_code`. The policy model is accessible at `https://huggingface.co/RYbWnlL0v8/llm_atp_model`

Table 1: Overall evaluation results across MiniF2F-test and ProofNet-test. All baseline results evaluated by us are marked with an asterisk (*), while the remaining results are directly imported from the original papers.

| Method | Model Size | Budget | MiniF2F-test | ProofNet-test |
|---|---|---|---|---|
| Llemma (Azerbayev et al., 2023b) | 7B | $1 \times 32 \times 100$ | 26.23% | - |
| ReProver (Yang et al., 2023) | 229M | K=1 | 26.50% | 13.80% |
| Lean-STaR (Lin et al., 2024) | 7B | $64 \times 1 \times 50$ | 46.30% | - |
| InternLM2.5-StepProver(Wu et al., 2024) | 7B | $1 \times 32 \times 600$ | $47.3\% \pm 1.1\%$ | $19.89\% \pm 0.68\%$ * |
| BFS-Prover(Xin et al., 2025) | 7B | $1 \times 2 \times 600$ | $55.49\% \pm 0.61\%$ * | $12.37\% \pm 1.44\%$ * |
| Ours (fixed beam size) | 7B | $1 \times 4 \times 600$ | $56.56\% \pm 1.13\%$ | $13.01\% \pm 0.79\%$ |
|  |  | $1 \times 8 \times 600$ | $\mathbf{59.51}\% \pm \mathbf{0.79}\%$ | $17.53\% \pm 0.26\%$ |
|  |  | $1 \times 16 \times 600$ | $59.02\% \pm 1.13\%$ | $19.35\% \pm 0.96\%$ |
|  |  | $1 \times 32 \times 600$ | $57.05\% \pm 0.4\%$ | $\mathbf{20.75}\% \pm \mathbf{0.94}\%$ |
| Ours (adaptive beam size) | 7B | $K = 1, E = 600$ | $\mathbf{60.74}\% \pm \mathbf{0.88}\%$ | $\mathbf{21.18}\% \pm \mathbf{0.55}\%$ |

model to explore a broader set tactics in the hope of reaching intermediate proof states similar to those encountered during training.

In tree search, a smaller beam size naturally makes the entire tree taller and thinner, resulting in longer average search depths (cf. Appendix A.8). Moreover, if the first expansion results are suboptimal, the subsequent search process may not reach proof finish at all. Therefore, when restricted to *Pass@1*, a beam size of 2 leads to larger variance, often requiring an increase in $K$ within the budget to reduce the impact of randomness. This small beam size scenario is suitable when, during tree search, the policy model encounters a state similar to one it has learned before and can confidently select a proof step which likely leads to proof finish. Conversely, a larger beam size makes the tree structure shallower and wider, reducing the average search depth but potentially introducing many useless proof steps into the tree, lowering the search efficiency. In addition, we conduct several supplementary experiments and identify significant issues in the function scoring of current tree search methods. A detailed discussion is provided in Appendix A.6.

During the data synthesis process, we observe that constrained decoding is necessary in a substantial number of cases, even though it may reduce the efficiency of distillation to some extent. A typical scenario we identify is when the model employs the `have` tactic to add a new hypothesis to the current goal, but the generated outputs differ only in the naming of the hypothesis. By enforcing the model to attempt different tactics, we largely mitigate this issue and introduce diversity.

## 5 RELATED WORK

**Automated theorem proving.** ATP has been a classic research area which is historically dominated by symbolic approaches (Pastre, 1993; Schürmann & Pfenning, 1998; Leino, 2013). In recent years, the rise of LLMs has spurred renewed interest. One representative work is GPT-f (Polu & Sutskever, 2020), which uses the Expert Iteration approach to train the LLM for the next proof step generation based on the current state, combined with tree search methods to prove theorems. Subsequent tree search methods have followed similar ideas, e.g., Polu et al. (2022); Lample et al. (2022); Lin et al. (2024). Additionally, some work improves tree search tailored in ATP. For example, Wu et al. (2024) uses the DPO algorithm to train an auxiliary value network to evaluate intermediate proof states, avoiding the issue of directly using raw beam search probabilities as scoring signals.

In contrast, whole-proof generation methods (Xin et al., 2024b; Lin et al., 2025; Zhang et al., 2025; Wang et al., 2025a) aim to produce an entire proof in a single forward pass from the initial goal. While these methods do not leverage intermediate proof states, they are typically more efficient and allow for multiple rollouts to increase the likelihood of success. A notable example is DeepSeek-Prover-V1.5 (Xin et al., 2024b), which enhances performance by inserting natural language chain-of-thought (CoT) commentary into Lean 4 proofs during inference. Additionally, MA-LoT (Wang et al., 2025b) introduces a multi-agent architecture where separate models are responsible for analyzing and revising incorrect proofs, enabling iterative refinement toward a correct solution.

**Data synthesis for theorem proving.** One of the major bottlenecks in LLM-based automated theorem proving remains the lack of high-quality training data, particularly for Lean 4, a relatively new language with limited existing formalized data. Existing data generation methods can be broadly categorized into two groups.

The first category leverages LLMs to generate Lean 4 problems by building on the abundance of natural language mathematics. These approaches are generally referred to as *autoformalization*. However, current efforts primarily focus on autoformalizing problem *statements*, with less attention given to the autoformalization of proof processes (Wang et al., 2025a; Lu et al., 2024).

The second category bootstraps from a small set of existing Lean 4 problems to generate new ones through iterative synthesis. These methods typically involve a two-model framework: one model generates new conjectures, and the other attempts to prove them, enabling co-evolution of data and model capabilities. For instance, STP (Dong & Ma, 2025) uses problems from the Lean-Workbook dataset (Ying et al., 2024) as seeds. A conjecturer proposes conjectures related to the original problems, while a prover attempts to validate them—allowing both the training set and the model to be refined through this iterative process.

# 6 LIMITATIONS

**Limited adaptivity in beam size selection.** As discussed in Section 4.3, beam size is a critical factor in the effectiveness of tree search. While our proposed strategy provides a simple and effective means of dynamically adjusting the beam size, a more principled solution would involve training a dedicated model that, given a current proof state, predicts the optimal beam size for the policy model to use. Ideally, such a model would offer fully adaptive control during search. However, in practice, we found this approach challenging to implement. The primary difficulty lies in collecting sufficient high-quality training data for learning beam size prediction. Moreover, it is difficult for such a model to operate independently of the policy model, since the optimal beam size can be highly dependent on the specific behavior of the underlying policy model. As a result, we opted for a simple adaptive beam size decay strategy that complements our data synthesis framework. Exploring how to build more sophisticated, model-agnostic beam selection mechanisms, potentially via auxiliary learning signals or joint training, remains an open direction for future work.

**Potentially suboptimal scoring function.** Most current tree search methods based on best-first search apply the scoring function directly to the tactic or action, rather than to the resulting proof state. We argue that a more general and expressive scoring function should evaluate the entire transformation path from the root state $s_0$ to the current state $s_i$, i.e.,

$$\text{score}(s_i) = f\big((s_0, t_0, s_1), (s_1, t_1, s_2), \ldots, (s_{i-1}, t_{i-1}, s_i)\big).$$

Since this formulation may be too complex to implement in practice, we adopt a simplified approximation inspired by first-order Markov processes, assuming $\text{score}(s_i) = f(s_{i-1}, t_{i-1}, s_i)$. This structure provides two key advantages: (1) it allows for immediate pruning of low-quality transformations during search, saving computational budget; and (2) it supports path-level quality estimation via aggregation, e.g., by computing $\sum_{n=1}^{j} \text{score}(s_n)$ or $\prod_{n=1}^{j} \text{score}(s_n)$ across the trajectory. Nevertheless, further exploration is needed to design more expressive, yet computationally feasible, scoring functions that better capture long-range dependencies in proof trajectories.

**Dependence on the seed dataset.** A fundamental limitation of our data synthesis framework is its dependence on the seed dataset. The synthesized data is effectively constrained to the local neighborhood of the seed problems in the sample space. As a result, if the target problem lies far from the seed set in terms of distribution or difficulty, our method may struggle to generate useful intermediate states or transformations. In other words, under our current formulation, producing examples that are significantly more difficult than those in the seed set remains an open challenge.

# 7 CONCLUSIONS

In this paper, we have presented a data synthesis method and an adaptive tree search pruning strategy for automated theorem proving in Lean 4. Our proposed approach enables large-scale generation of intermediate proof states and transformation steps without relying on ground-truth solutions, while maintaining diversity in the synthetic data. Combining with the adaptive beam size strategy, our method effectively balances exploration and exploitation throughout the tree search process. Together with the developed DoBeVi Lean 4 interaction tool, we expect our contributions will support and inspire further research within the ATP community, which is burgeoning but under development.

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

# A  APPENDIX

## A.1  LLM USAGE

In this work, we make limited use of LLMs as auxiliary tools for writing and figure preparation. Specifically, LLMs assist in refining text clarity, improving grammar, and correcting minor wording issues. For certain figures, we also generate initial plotting scripts with the help of LLMs, which are subsequently revised and finalized by the authors. Importantly, all research ideas, theoretical developments, algorithmic designs, and experimental analyses are conceived and carried out independently by the authors without reliance on LLMs.

## A.2  PROMPTS USED FOR POLICY MODEL

For the policy model, we use the similar prompt template as in GPT-f (Polu & Sutskever, 2020), where the prompt only describes the current proof state without introducing any additional information. The template and an example are shown below:

> *Template:*
>
> [GOAL]
> $<$state$>$
> [PROOFSTEP]
>
>
> *Example:* (aime_1983_p1 in MiniF2F)
>
> [GOAL]
> x y z w : $\mathbb{N}$
> ht : $1 < x \land 1 < y \land 1 < z$
> hw : $0 \le w$
> h0 : Real.log $\uparrow$w / Real.log $\uparrow$x = 24
> h1 : Real.log $\uparrow$w / Real.log $\uparrow$y = 40
> h2 : Real.log $\uparrow$w / Real.log ($\uparrow$x * $\uparrow$y * $\uparrow$z) = 12
> $\vdash$ Real.log $\uparrow$w / Real.log $\uparrow$z = 60
> [PROOFSTEP]

## A.3  CASE STUDY

Below we present two example proofs generated by our model on MiniF2F and ProofNet, respectively.

> Example: amc12a_2009_p6 in MiniF2F
>
> ```
> theorem amc12a_2009_p6
>   (m n p q : ℝ) ℝ
>   (h0 : p = 2 ^ m)
>   (h1 : q = 3 ^ n) :
>   p^(2 * n) * (q^m) = 12^(m * n) := by
>     subst_vars
>     simp [mul_comm, Real.rpow_mul]
>     have h0 : (0 : ℝ) < 2 := by norm_num
>     have h1 : (0 : ℝ) < 3 := by norm_num
>     field_simp [Real.rpow_def_of_pos, h0, h1]
>     rw [← Real.exp_add]
>     rw [mul_comm]
>     rw [
> ```

```
        show (n * 2 * (Real.log 2 * m) + Real.log 3 * n * m) =
    (Real.log 12 * m * n) by
        have h3 : Real.log 12 = Real.log (2 ^ 2 * 3) := by norm_num
        rw [h3]
        field_simp [Real.log_mul, Real.log_rpow, mul_assoc]
        ring_nf
    ]
```

---

Example: Rudin_exercise_4_11a in ProofNet

```
theorem exercise_4_11a
    {X : Type*} [MetricSpace X]
    {Y : Type*} [MetricSpace Y]
    (f : X → Y) (hf : UniformContinuous f)
    (x : ℕ → X) (hx : CauchySeq x) :
    CauchySeq (λ n => f (x n)) :=
        rw [Metric.cauchySeq_iff] at hx ⊢,
        intro ε hε,
        obtain ⟨δ, hδ, hδ'⟩ := Metric.uniformContinuous_iff.mp hf ε
    hε,
        obtain ⟨ N, hN ⟩ := hx δ hδ,
        exact ⟨ N, fun m hm n hn => hδ' <| hN _ hm _ hn ⟩
```

## A.4 OUR DOBEVI REPL

To enable better interaction with Lean 4 prover and visualization of the search process, we develop Dojo-BeamSearch-Visualization (DoBeVi) REPL, a tool built upon a simplified and customized version of LeanDojo (Yang et al., 2023). The tool consists of two main components: *interactive theorem proving environment* and *visualization module*.

### A.4.1 INTERACTIVE THEOREM PROVING ENVIRONMENT

In the component of interactive theorem proving environment, we present a streamlined adaptation of the LeanDojo framework, establishing a lightweight but robust foundation for distributed interactive theorem proving. Our implementation retains two core functionalities from LeanDojo:

- AST extraction and semantic analysis of target Lean files with precise theorem declaration identification.
- Core interactive logic execution via the lean_dojo_repl custom tactic.

In our DoBeVi REPL refines the architecture by removing three categories of non-essential components from LeanDojo:

- Partial initialization workflows.
- Premise and dependency tracking modules.
- Secondary data processing components.

Then we add the following new useful features for better interaction with the prover:

- Proof state tracking using globally unique identifiers (excluding ProofFinish nodes).
- Decoupled session management architecture enabling parallel theorem processing.
- Resilient recovery mechanisms for:
    - Tactic execution timeouts.
    - Lean process failures.

After that, the DoBeVi provides an optimized architecture that reduces runtime overhead while maintaining stable interfaces for integrating model-based theorem proving components.

### A.4.2 VISUALIZATION OF THE SEARCH TREE

The visualization module serves to clearly illustrate the entire proof search process by graphically representing the evolving search tree structure. In this module, each proof state encountered during the search is abstracted as a node in a graph, and each proof step applied to transition between states is represented as a directed edge. To better reflect the semantics of the search process, nodes are categorized into three types:

- ProofFinished node: Indicates that the proof has been successfully completed at this node.
- Error node: Represents a failed search node due to errors such as Lean syntax errors, timeouts, or unexpected crashes of the Lean process.
- Open node: Denotes an intermediate state that can still be expanded during the search.

Similarly, edges are divided into two categories:

- Tree edge: Connects to a previously unseen state, resulting in the creation of a new node. Tree edges are essential for expanding the search space and discovering new proof paths.
- Back edge: Leads to a state that has already been visited earlier in the search process. Back edges help identify convergence and redundancy in the search, enabling cycle detection and pruning.

Although the structure is referred to as a "search tree", it is, strictly speaking, a directed acyclic graph (DAG). This distinction stems from two key characteristics:

- Non-tree structure: Different but semantically similar proof steps may both lead from the same current state to the same next state, resulting in multiple paths converging to a single node. The property that there are no loops inside the tree is violated.
- Acyclic design: Cycles in the search path (e.g., $s_1 \rightarrow s_2 \rightarrow \cdots \rightarrow s_1$) are explicitly detected and pruned in advance, as we consider such loops to be semantically meaningless and computationally redundant.

This visualization module is part of the DoBeVi system, which is organized in a modular fashion. The system consists of the following four components:

- Search tree module: Defines the core data structures, including Node, Edge, and the overall Tree layout.
- Tactic generation module: Interfaces with LLMs. Given the current state as input, it returns a set of candidate proof steps along with their associated scores. This module is extensible and allows users to integrate custom models as needed.
- Search strategy module: Specifies the logic for node expansion. We provide several default strategies such as Best-First Search, which always expands the node with the highest current score that has not yet been explored. Users are free to implement additional strategies tailored to their tasks.
- Visualization module: Responsible for rendering the entire search tree graphically, enabling an intuitive understanding of the search process and state transitions.

Figure 3 presents the visualization result of applying the Best-First Search strategy to the problem amc12_2000_p6 from the MiniF2F dataset. Node IDs increment from 0, indicating the order in which nodes were expanded during the search. In this example, the search sequentially expanded nodes with IDs 0, 3, 11, 14, 16, 19, and 21, ultimately completing the proof successfully after the seventh expansion.

In summary, this module visualizes the search tree to record the entire search process, making it easier to review after execution and to inspire the design of better search and pruning strategies.

### A.5 DETAILED EXPERIMENTAL SETTINGS

In our experiments, similar to most tree search methods, we separate the tree search and model generation components to improve overall evaluation efficiency. We refer to the implementation

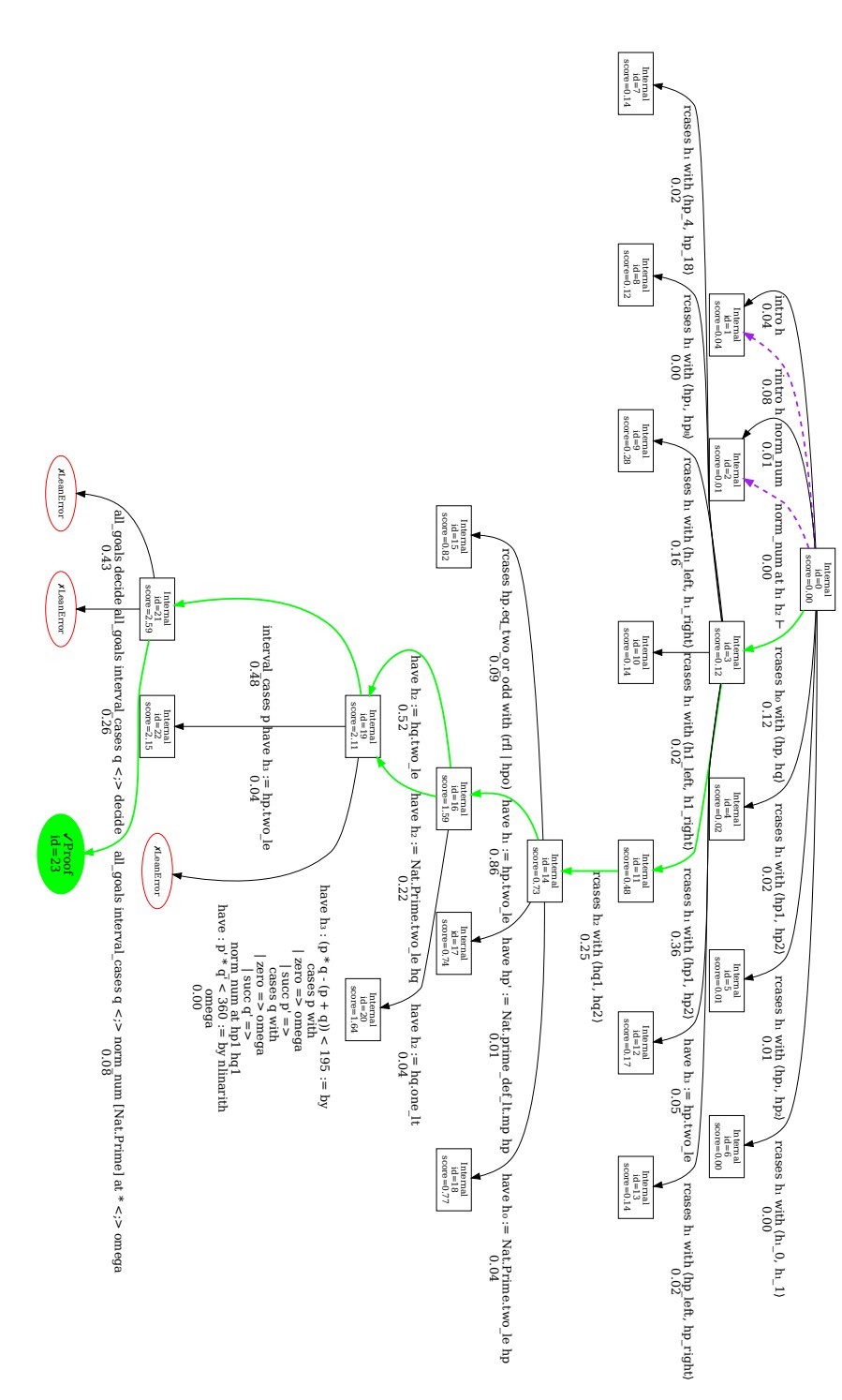

Figure 3: A example of search tree(amc12_2000_p6 in MiniF2F). Rectangular nodes represent `Open Nodes`, red elliptical nodes indicate `Error Nodes`, and green nodes denote `ProofFinished Nodes`. Edges are labeled with the applied tactic and its associated beam probability (`beam_prob`). Each node is annotated with its unique id and current score.

in ReProver (Yang et al., 2023) and use Python's Ray [4] and vLLM (Kwon et al., 2023) libraries to implement the system. We launch multiple processes for tree search, where each process pulls unprocessed theorems from a queue and uses LeanDojo as the interaction tool between Python and the Lean 4 prover. When fetching proof steps, multiple tree search workers share several asynchronous LLM engines launched by vLLM, with one LLM instance deployed per GPU. In practice, we use dozens of tree search processes to maximize GPU utilization. For each search, we set a global timeout of 1,800 seconds; if the search time exceeds this value, the search process is forcibly terminated. For each proof step, we impose a timeout of 20 seconds. While there are occasional cases - such as when using `aesop` - where longer execution times may be required, we consider any proof step exceeding this threshold as a timeout. This constraint is intended to prevent individual proof steps from disproportionately consuming the overall time budget.

During tree search, we use the REPL functionality provided by DoBeVi (Appendix A.4). When executing proof steps, errors can broadly be categorized into two types: (1) syntax errors or similar issues that directly cause the Lean 4 prover to return an error, which are usually caught quickly with an error message; and (2) timeout errors. To prevent infinite loops caused by certain tactics, we set a timeout for tactic execution. If such a timeout occurs, the Lean 4 prover becomes unable to execute new proof steps, forcing the entire proof process to stop. To handle this, we save the proof context before executing each tactic; if a timeout occurs, we restart the Lean REPL environment and restore the context. This avoids cases where a proof that should have succeeded with tree search ultimately fails just because a faulty tactic caused the Lean 4 prover to hang.

In addition, based on observations of certain failure cases during tree search, we identified an important phenomenon: erroneous proof steps tend to appear in batches. Specifically, when performing beam search from an proof state, if several consecutive proof steps are incorrect, it is highly likely that the remaining proof steps in the beam will also be incorrect. This pattern may stem from the fact that the current proof state represents an out-of-distribution input for the policy model, making it difficult for the model to generate valid Lean 4 code. Under such circumstances, these erroneous proof steps can substantially consume the global timeout allocated for the tree search, leading to inefficient exploration. To mitigate this issue, we adopt an early termination strategy: during expansion at a given state, if the number of incorrect child nodes exceeds a predefined threshold (usually $0.5\times$ beam size), we immediately discard the current node from further search. We found that this pruning technique did not degrade overall proving performance while significantly reducing computational time.

Regarding the choice of hyperparameters in the adaptive beam size strategy, we set $B_{\max} = 16$, $B_{\min} = 4$, and $\lambda = 15$ for MiniF2F, and $B_{\max} = 48$, $B_{\min} = 24$, and $\lambda = 2$ for ProofNet.

### A.6 Unsuccessful attempts and discussions

In our experiments, we adopt a standard formulation for the scoring function, which evaluates the tactic $t_j$ rather than the new state $s_{i+1}^j$. As shown in Equation 1, the effectiveness of this scoring function primarily stems from the fact that a higher value of $P(t_j \mid s_i)$ indicates that the policy model is more likely to have encountered proof states similar to $s_i$ during training. As a result, it can, to some extent, prioritize proof steps that are more helpful for completing the proof. However, we argue that this scoring function still suffers from significant issues.

**Separating tactics and premises.** In our early experiments, we explored a more straightforward separation of tactics and premises. Specifically, we adopted a two-model design: one model directly outputs a valid tactic to serve as the starting token of the code, while a second model, analogous to the policy model, completes the required premises. However, this approach yielded unsatisfactory results. We frequently observed that the second model either failed to generate complete premises or produced outputs that violated the Lean 4 grammar. A key challenge lies in the highly imbalanced data distribution for a model that generates only tactics without premises. To quantify this imbalance, we analyzed the STP dataset and computed the tactic usage statistics, presented in Appendix A.7. For example, tactics like `rw` and `have` appear in nearly every proof context, leading the model to overwhelmingly favor these high-frequency tactics. In contrast, the policy model must guide the proof state toward completion, which does not always align with simply selecting the most common tactics. This mismatch introduces conflicting objectives that ultimately degrade performance. Therefore,

---

[4] https://github.com/ray-project/ray

we conclude that forcibly separating the generation of tactics and premises is infeasible. Instead, it may be more effective to adopt external augmentation strategies, such as the retrieval-augmented generation (RAG) method in Yang et al. (2023) or the lemma library construction approach in Wang et al. (2023).

**An ineffective scoring function.** One major drawback of tree search methods compared to whole-proof methods lies in the complexity of the neural-symbolic system: it involves many interacting components, making it difficult to diagnose failure modes when performance is suboptimal. In some cases, the coupling between components may even be inherently unavoidable. As discussed in the main text, we identify several key factors in tree search methods that have a non-negligible impact on performance:

- **Policy model.** As the actor in the system, the policy model is clearly the most critical determinant of overall performance.

- **Scoring function (sometimes value network).** The scoring function determines the order in which nodes are expanded within the tree structure. In some cases, a value network may be used for scoring instead of a rule-based method.

- **Beam size.** The beam size represents a special type of budget. As discussed in Section 4.3, increasing the beam size does not necessarily lead to better results. On the contrary, for certain settings, a small beam size (such as 4) can sometimes unlock greater potential from the policy model. Moreover, the empirically optimal beam size varies across benchmarks and is not a fixed value.

- **Other budget-related factors.** These include parameters like $K$, $E$, and the timeout threshold. Compared to the previous factors, these are relatively straightforward: given fixed settings for other components, larger budgets generally increase the likelihood of achieving better results.

In our early experiments, we hypothesized that the problem setting resembled reinforcement learning, and thus sought inspiration from algorithms such as GRPO Shao et al. (2024). We aimed to avoid introducing a separate value network, instead using the average performance of a set of actions generated by the actor as a scoring criterion. Concretely, for any intermediate proof state $s$, we computed the following score:

$$\text{score}(s) = \frac{1}{B} \sum_{i=1}^{B} \log P_\theta(t_i \mid s) \tag{3}$$

Here, $\theta$ denotes the policy model's parameters, and $P_\theta(t_i \mid s)$ represents the joint probability of the $i$-th beam. The intuition behind this scoring function is that, on an average sense, higher beam probabilities suggest that the policy model is more likely to have encountered similar examples during training. Therefore, the current proof state $s$ is more likely to lie on a path leading to proof completion.

However, subsequent experiments indicate that the evaluation results obtained using this scoring function are nearly indistinguishable from those derived using Equation 1, suggesting that this approach may not be particularly effective within the context of tree search methods.

**Balancing exploration and exploitation.** In search process, a common optimization strategy is to balance exploration and exploitation, with classical algorithms such as MCTS following this principle. In the ATP context, a number of methods with similar ideas have also been proposed, for instance the length-normalized BFS introduced in Xin et al. (2025). However, in our experiments, we find that these approaches show no clear difference from naive BFS in terms of evaluation results.

Applying the standard MCTS framework to ATP presents several significant challenges. First, the selection step in MCTS relies on the Upper Confidence Bound (UCB), which requires access to the parent node's visit count. Although the ATP search process is often described as a "tree search," the underlying structure is in fact a directed acyclic graph (DAG), since the same intermediate proof state can be reached through multiple derivation paths. This makes it ambiguous, or even impossible, to assign a unique parent visit count to each node. Second, the classical simulation step in MCTS is impractical for ATP. Once a simulation reaches a successful proof state, the proof is complete and no

further search is necessary. Moreover, simulations consume interactions with the Lean server step by step, which also exhausts the allocated budget. As a result, applying the classical simulation of MCTS to ATP effectively reduces to a form of best-first search. In practice, we experiment with modified versions of MCTS to address these issues. On the MiniF2F-test benchmark under the $1 \times 8 \times 600$ setting, the modified MCTS achieves an average *pass@1* of $56.15\%$, which is slightly lower than that of naive BFS.

We also explore length-normalized BFS, which normalizes by the current depth of the tree node in an attempt to encourage the expansion of deeper nodes rather than restricting exploration to shallow levels. However, we find that its generalization ability is limited, and when combined with our policy model, it has virtually no effect to the tree search process.

**Dynamic beam size via *top_p* strategy.** Another adaptive beam search strategy we explored is inspired by the *top_p* (nucleus sampling) strategy commonly used in decoding. Specifically, after normalizing the joint probabilities of each beam, we sort the beams by their joint probabilities and then filter out beams with lower probabilities according to the *top_p* threshold.

**Conclusion.** Considering all of the strategies we experimented with, we conclude that the scoring function (Equation 1) used in current tree search methods still suffers from significant shortcomings. As a result, the experimental outcomes of these various approaches showed little impact on performance. Further analysis for scoring function please refer to A.12.

### A.7  TACTIC USAGE STATISTICS ON THE STP DATASET

We performed a comprehensive scan of all tactics used across the entire STP dataset, identifying a total of 173 unique tactics. By applying a threshold of $\mathcal{P} = 0.999$ under the *top_p* strategy to exclude infrequently used tactics, we obtained a final set of 60 common tactics. The frequency distribution of these tactics is shown in Figure 4.

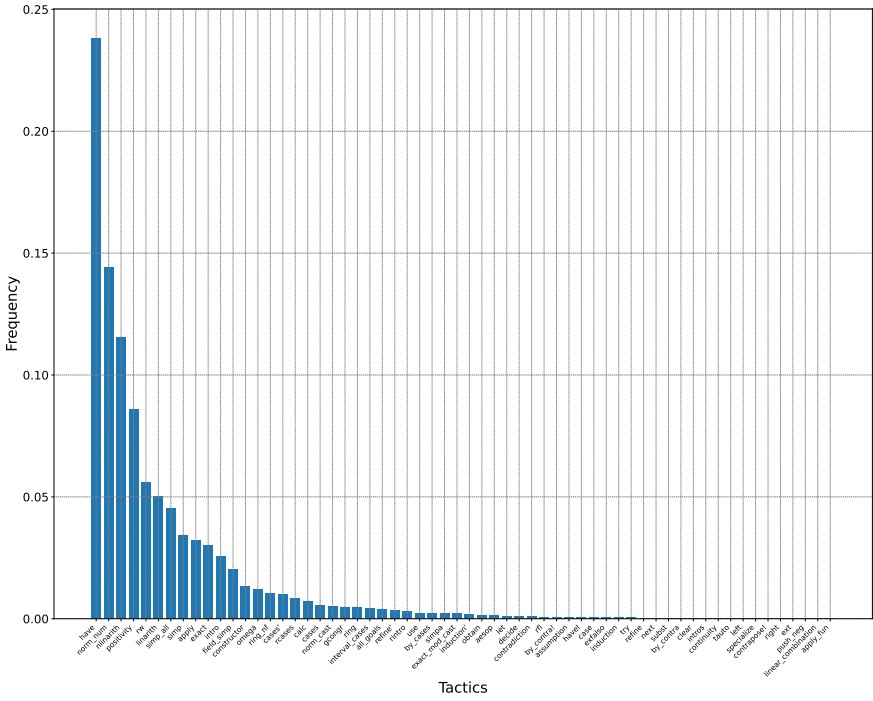

Figure 4: Distribution of the top 60 most frequent tactics in the STP dataset.

## A.8  SEARCH DEPTH STATISTICS ON THE MINIF2F BENCHMARK

We conducted experiments using our policy model on the MiniF2F benchmark with varying beam sizes during search. The results, shown in Figure 5, indicate that under a fixed maximum number of expansions, smaller beam sizes tend to produce proving paths with greater search depth.

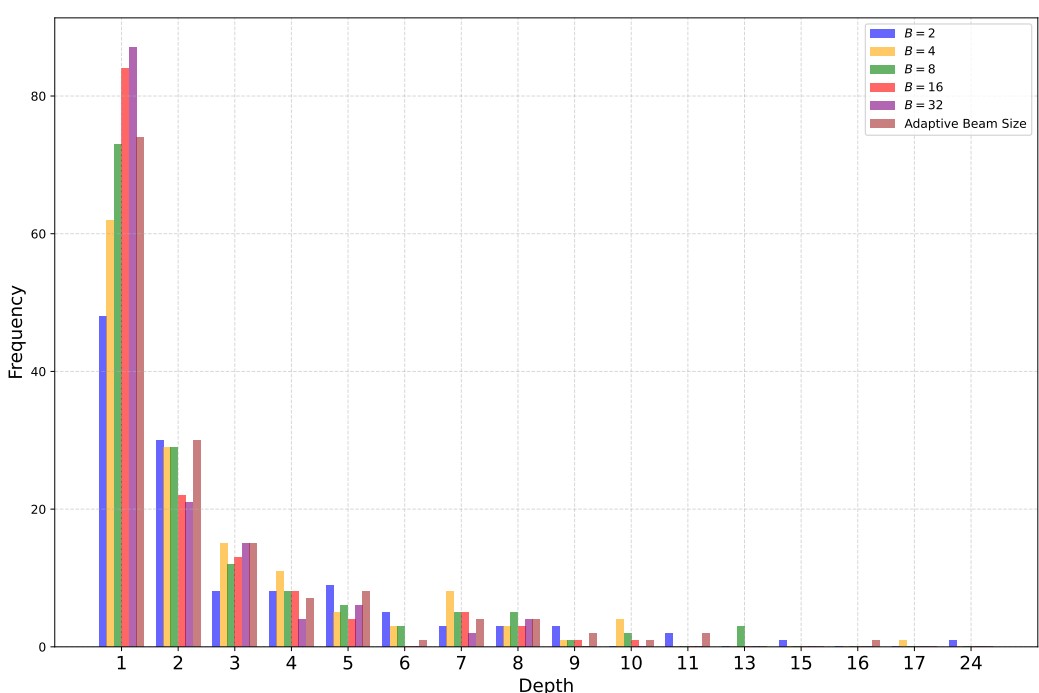

Figure 5: Frequency distribution of search depths of proving paths generated by our policy model on the MiniF2F benchmark, with the maximum number of expansions fixed at 600 ($E = 600$).

## A.9  BASIC FUNCTIONALITY OF THE EXAMPLE TACTICS

Below we summarize the Lean 4 tactics used in our examples, together with minimal one-line illustrations. Each illustration uses comments to indicate the context and the effect of the tactic.

- **rw** (rewrite): Rewrites the goal using a known equality.

  ```
  -- Context: h : a = b, goal: f a = c
  rw [h]
  -- After: goal becomes f b = c
  ```

- **have**: Introduces an intermediate fact for later use.

  ```
  -- Before: no lemma about x + y
  have h1 : x + y = y + x := by apply add_comm
  -- After: h1 is available as an additional assumption
  ```

- **rcases**: Decomposes a structured hypothesis (e.g., an existential or a conjunction).

  ```
  -- Context: h : ∃ n, n > 0 ∧ n < 5
  rcases h with ⟨n, hn_pos, hn_lt5⟩
  -- After: n : Nat, hn_pos : n > 0, hn_lt5 : n < 5
  ```

- **apply**: Matches the goal with the conclusion of a lemma, turning it into its premises.

  ```
  -- Context: goal is x ≤ z, lemma le_trans : x ≤ y → y ≤ z → x ≤
       z
  ```

```
        apply le_trans
        -- After: two subgoals: x ≤ y  and  y ≤ z
```

- **field_simp**: Simplifies expressions in fields by clearing denominators.

```
        -- Context: goal contains a rational expression, assume h : x ≠ 0
        field_simp [h]
        -- After: denominators are cleared, e.g., (a / x) * x becomes a
```

- **nlinarith**: Automatically solves nonlinear arithmetic goals.

```
        -- Context: hx : x ≥ 0, hy : y ≥ 0, goal: x * y ≥ 0
        nlinarith
        -- After: goal is discharged automatically
```

- **rfl**: Closes goals that are definitionally equal.

```
        -- Context: goal: (x + 1) - 1 = x
        rfl
        -- After: goal closed because both sides reduce to the same
            expression
```

## A.10 Ablation Study and Further Analysis of Data Synthesis and Adaptive Beam Size Strategy

### A.10.1 Data Synthesis

We provide additional clarification regarding the roles of the three hyperparameters used in our data synthesis procedure.

$\alpha$ controls depth-based exploitation-exploration. This hyperparameter controls the balance between exploitation and exploration with respect to tree depth in the search tree. Given a beam size of $B$, we retain only $\alpha B$ branches at each step. During data synthesis, we typically set $B = 32$ to allow the policy model to generate tactics to which it assigns relatively high confidence. However, such a large beam size makes the search tree excessively wide, making it difficult to obtain new states that differ significantly from the original state after several proof steps. This effect is also visible during evaluation (e.g., on MiniF2F, where the policy model is relatively familiar with the domain, a beam size of 32 is not optimal.). By keeping only $\alpha B$ branches, we obtain more nodes from deeper parts of the tree, improving exploration.

$\beta$ controls branch-wise exploitation-exploration. This hyperparameter regulates the trade-off across different branches within beam search. Naive data synthesis tends to overemphasize exploitation, while pure constrained decoding risks pushing the model into excessively exploratory behavior, forgetting solving the problem. To avoid both extremes, we include the top-$\beta$ most likely branches (ensuring exploitation) and randomly sample additional branches from the remaining candidates (ensuring exploration). This mixture maintains diversity in the synthesized data while preventing the policy model from drifting too far from problem-solving behavior.

$\gamma$ is responsible for budget decay for easy problems. This hyperparameter helps terminate the search early for problems that are too easy. In the STP dataset, some problems require only a few steps to reach a proof-finished state. Unlike evaluation (where search terminates as soon as one proof is found), data synthesis aims to discover multiple distinct proof-finished paths. However, allocating the same search budget to every seed problem is not appropriate. Therefore, each time a new proof-finished path is discovered, the remaining search budget is multiplied by a decay factor $\gamma < 1$. Easy problems naturally yield multiple finished paths, causing their budgets to decay quickly and terminating the search earlier. A smaller $\gamma < 1$ results in faster decay.

Furthermore, we conduct a small-scale ablation study using the same policy model (BFS-Prover) and the same seed dataset (STP) as in our proposed method, in order to validate its effectiveness. During data synthesis, we employ greedy decoding to generate proof steps with BFS-Prover. Starting from approximately 60,000 randomly sampled problems as seeds, we obtain about 1.04 million state–tactic pairs as the raw synthesized data. We then remove samples that produce errors when validated by the Lean prover and apply the same data contamination checks as in our method. After filtering,

roughly 580,000 samples remain, forming the final greedy-decoding-based synthesized dataset used for training.

To ensure a fair comparison, we randomly sample an equal-sized subset of 580,000 examples from our own synthesized dataset. Both datasets are used to fine-tune the same base model, Qwen2.5-Math-7B, and are evaluated on MiniF2F-Test with identical budgets and experimental settings. The results are shown in Table 2.

Table 2: Comparison of data synthesis strategies between greedy decoding and our data synthesis method.

| Budget | Greedy Decoding | Ours |
|---|---|---|
| $1 \times 8 \times 600$ | 51.64% | 55.74% |
| $1 \times 16 \times 600$ | 50.82% | 56.15% |
| $1 \times 32 \times 600$ | 50.00% | 54.92% |
| $K = 1, E = 600$, adaptive beam size | 52.87% | 56.15% |

These results show that our data synthesis method can improve the diversity of generated training examples. This diversity helps the model better understand the applicable contexts of different tactics and enables it to explore a broader set of solution strategies during problem solving. In contrast, data synthesized with greedy decoding tends to overemphasize exploitation. This issue is especially pronounced when the policy model has been trained through expert iteration, as the model often produces highly similar proof steps when encountering the same state. When such homogeneous data is used for training, the resulting policy model tends to generate proof steps that are very similar to one another, which limits overall performance.

### A.10.2 ADAPTIVE BEAM SIZE STRATEGY

We additionally evaluate the adaptive beam size strategy on MiniF2F using our model as the policy model. In this study, we vary the three hyperparameters associated with adaptive beam size and evaluate each configuration twice. The results are summarized in Table 3. Overall, the results indicate that as long as $B_{\max}$ and $B_{\min}$ are chosen within a reasonable range, the performance is relatively robust to the specific choices of the three hyperparameters.

Table 3: Evaluation of adaptive beam-size decay hyperparameters on MiniF2F. Each configuration is run twice; we report the average Pass@1.

| $B_{\max}$ | $B_{\min}$ | $\lambda$ | Average Pass@1 |
|---|---|---|---|
| 16 | 4 | 15 | 60.74% |
| 16 | 4 | 5 | 60.45% |
| 16 | 4 | 30 | 60.04% |
| 32 | 16 | 15 | 59.22% |
| 32 | 4 | 15 | 60.04% |
| 16 | 2 | 15 | 60.04% |
| 8 | 2 | 15 | 59.22% |

Based on these findings, we summarize the practical guidelines for selecting hyperparameters in the adaptive beam size strategy as follows:

$B_{\max}$: Given a fixed evaluation benchmark, let $B^*$ denote the optimal fixed beam size. We recommend setting $B_{\max}$ slightly larger than $B^*$, but generally not exceeding $2B^*$. This ensures sufficient exploration without unnecessarily enlarging the search space.

$B_{\min}$: We recommend values of 2 or 4. Empirically, we observe no substantial difference between these choices. Because $B_{\min}$ governs the beam size in deeper layers of the search tree, using a relatively small value can slightly improve Pass@1 by encouraging deeper exploration.

## A.11 APPLICABILITY OF ADAPTIVE BEAM SIZE STRATEGY TO DIFFERENT PROBLEM TYPES

To further examine whether the adaptive beam size strategy behaves differently across problem types, we conduct an additional set of experiments using the official category labels from MiniF2F (e.g., original competition source or mathematical subfield: *aime*, *algebra*, *amc*, *induction*, *numbertheory*, etc.; see Zheng et al. (2021)). For each category, we perform 15 independent evaluations on the MiniF2F-Test split for both the fixed-beam and adaptive-beam strategies. Specifically, we run five evaluations for each fixed beam size $B = 8, 16, 32$, and 15 evaluations for the adaptive beam size strategy. We then compute success rates under each category using two metrics below.

*Cumulative solved.* This metric measures the proportion of problems that are solved at least once across all runs. For instance, if a category contains 100 problems and 82 of them are solved in at least one of the 15 runs, then the cumulative solved score is $82/100 = 0.82$.

*Overall solved.* This metric measures the average solve rate across all runs. It is computed as the total number of solved instances across all runs divided by the number of problems times the number of runs. For example, with 100 problems and 15 runs, if the total number of solved instances is 1,080, then the overall solved metric is $1080/(100 \times 15) = 0.72$.

The results are presented in Table 4 and Table 5. Across all subcategories of MiniF2F-Test, the adaptive beam size strategy consistently improves performance, rather than yielding gains concentrated in specific domains. As a general enhancement of the search procedure, its benefits appear broadly applicable across diverse mathematical problem types.

Table 4: Performance of **fixed beam size strategy** across MiniF2F-Test subcategories, evaluated over 15 runs.

| Subcategories | Cumulative Solved | Overall Solved |
|---|---|---|
| aime | 5/15 (33.33%) | 64/225 (28.44%) |
| algebra | 10/18 (55.56%) | 147/270 (54.44%) |
| amc | 21/45 (46.67%) | 245/675 (36.30%) |
| imo | 3/20 (15.00%) | 35/300 (11.67%) |
| induction | 5/8 (62.50%) | 74/120 (61.67%) |
| numbertheory | 4/8 (50.00%) | 53/120 (44.17%) |
| mathd_numbertheory | 51/60 (85.00%) | 701/900 (77.89%) |
| mathd_algebra | 60/70 (85.71%) | 813/1050 (77.43%) |

Table 5: Performance of **adaptive beam size strategy** across MiniF2F-Test subcategories, evaluated over 15 runs.

| Subcategories | Cumulative Solved | Overall Solved |
|---|---|---|
| aime | 5/15 (33.33%) | 70/225 (31.11%) |
| algebra | 10/18 (55.56%) | 149/270 (55.19%) |
| amc | 22/45 (48.89%) | 271/675 (40.15%) |
| imo | 4/20 (20.00%) | 46/300 (15.33%) |
| induction | 7/8 (87.50%) | 78/120 (65.00%) |
| numbertheory | 4/8 (50.00%) | 60/120 (50.00%) |
| mathd_numbertheory | 53/60 (88.32%) | 716/900 (79.56%) |
| mathd_algebra | 62/70 (88.57%) | 833/1050 (79.33%) |

## A.12 FURTHER ANALYSIS OF SCORING FUNCTION

To examine the effectiveness of different scoring functions, we select several representative methods and conduct additional experiments on the MiniF2F-Test split using our model. The experimental budget is set to $K = 1$, $B = 8$, and $E = 600$, with a global timeout of 1800 seconds per problem. The results are shown in Table 6.

From these results, we observe that the performance differences between scoring functions are small. Several normalization-based strategies that are theoretically appealing (such as depth normalization

Table 6: Comparison of representative scoring functions on MiniF2F-Test under fixed search budget.

| Scoring Function | Results |
|---|---|
| *log probability* | $59.9\% \pm 0.9\%$ |
| *log probability + depth normalization* ($\alpha = 0.5$) (Xin et al., 2025) | $59.5\% \pm 1.7\%$ |
| *log prob + length normalization* | $57.7\% \pm 1.7\%$ |
| Value network (InternLM2.5-StepProver-critic) (Wu et al., 2024) | $57.2\% \pm 1.0\%$ |

and length normalization) did not lead to the expected improvements. Similarly, using a value-network-based scoring function did not offer clear benefits under the current setup. For this reason, we prefer *log probability* as the default scoring function, as it provides more stable performance across problems.

To more concretely illustrate the limitations of existing scoring functions in current tree-search methods, we take the example of solving the MiniF2F-Test *mathd_algebra_598* as an example, evaluated under the scoring function *log probability + depth normalization*. The initial state of the theorem is:

```
a b c d : ℝ
 h₁ : 4 ^ a = 5
 h₂ : 5 ^ b = 6
 h₃ : 6 ^ c = 7
 h₄ : 7 ^ d = 8
 ⊢ a * b * c * d = 3 / 2
```

During the search process, each state's score is computed as: score = parent_score + current_score, where current_score is obtained from the scoring function (*log probability + depth normalization*), and parent_score is the accumulated score along the path. Using this method, the search is able to correctly produce the first eight proof steps:

```
have h₅ : (4 : ℝ) ^ a = 5 := by linarith
have h₆ : (5 : ℝ) ^ b = 6 := h₂
have h₇ : (6 : ℝ) ^ c = 7 := h₃
have h₈ : (7 : ℝ) ^ d = 8 := h₄
have h₉ : (4 : ℝ) > 0 := by norm_num
have h₁₀ : (5 : ℝ) > 0 := by norm_num
have h₁₁ : (6 : ℝ) > 0 := by norm_num
have h₁₂ : (7 : ℝ) > 0 := by norm_num
```

At the next expansion step, a correct proof step would be :

```
have h₁₃ : Real.log 4 > 0 := Real.log_pos (by norm_num)
```

However, the policy model assigns it a relatively low score $-9.29$, while three other incorrect proof steps receive scores around $-4$. As a result, the search prefers the incorrect steps and subsequently moves into an unproductive subtree. At this point, the search depth has already reached 8. Due to depth normalization, states at different depths are mapped to a similar scale, the search struggles to escape once it enters a wrong subtree, often consuming a large portion of the budget without making progress (i.e., solve the problem). Without depth normalization, the scores of such incorrect subtrees would decrease quickly as the depth grows, allowing the search to abandon them much earlier.

This example shows that depth normalization can have a negative impact in certain cases. However, this does not imply that depth normalization is always harmful. When the policy model assigns strong and consistent preferences to correct steps, depth normalization can help keep the search focused

on the correct proof trajectory. Our intention with this example is to illustrate that existing scoring functions in tree-search methods can have both positive and negative effects depending on the specific problem, and each approach carries its own strengths and limitations.

