# OpenReview forum: "LLM-based Automated Theorem Proving Hinges on Scalable Synthetic Data Generation"
_ICLR.cc/2026/Conference — Submitted to ICLR 2026_

### Official Review · Reviewer_yJ1K · 2025-10-30

**Soundness:** 3
**Presentation:** 2
**Contribution:** 2
**Rating:** 2
**Confidence:** 3

**Summary:**

This paper addresses automated theorem proving (ATP) in Lean 4 using large language models through a tree search approach. The authors propose two main contributions: (1) a proof state exploration method for synthetic data generation that enforces diversity by using constrained decoding over a curated set of 60 common tactics, and (2) an adaptive beam size strategy that starts with large beam sizes for exploration and gradually reduces them for exploitation. The method achieves 60.74% on MiniF2F and 21.18% on ProofNet under Pass@1, outperforming existing tree search baselines. The approach avoids expert iteration and relies on one-shot fine-tuning with approximately 20 million synthetic proof transitions.

**Strengths:**

- Well-mentioned Contributions
  - The proof-state exploration method with constrained decoding is clearly motivated and addresses a critical bottleneck in LLM-based ATP. Using constrained decoding to force diversity over 60 curated tactics. The constrained decoding mechanism for forcing tactic diversity is well-motivated and technically reasonable.
  - Adaptive beam size strategy, a simple linear decay function for beam size. The Equation 2 is simple and interpretable
  - DoBeVi tool: A useful engineering contribution but primarily a refactored version of LeanDojo with visualization.
- Clear identification of beam size importance: The analysis in Section 4.3 and Figure 5 clearly demonstrates how beam size affects tree structure and performance, which is an underappreciated factor in prior work
- Honest disclosure of limitations: Section 6 and Appendix A.6 transparently discuss failed attempts (MCTS, scoring functions, tactic/premise separation), which is valuable for the community
- Scalability: Generating 20M training samples and achieving one-shot fine-tuning without expert iteration is practically valuable

**Weaknesses:**

- The abstract lacks clarity: The phrase *"proof-state exploration approach for training data synthesis"* is vague; readers won't understand what this means without reading the full paper.
- The motivation flow feels disjointed — Section 1 jumps between concepts (tree search vs. whole-proof, data scarcity, EI inefficiency) without a clear narrative thread.
- Figure 1 is confusing: the example showing tactics like `rw`, `rcases`, `have` without explaining what these mean or why they matter is not helpful for a general ML audience (although some are mentioned later).
- Algorithm 1 has notation inconsistencies: it uses `s0, n` and then drops `n`, and mixes implementation details (`q.add`, `lean_prover`) with algorithmic logic.

- No comparison with whole-proof generation methods (e.g., *DeepSeek-Prover-V1.5*).
- No ablation studies isolating the contribution of
  (a) data synthesis vs.
  (b) adaptive beam size.
- No analysis of what types of problems benefit most from the approach.

- The scoring function is acknowledged to be flawed (Section 4.3, Appendix A.6: *"significant issues," "nearly indistinguishable"* from baseline).
- Multiple exploration strategies failed (*MCTS, length-normalized BFS, top-p*).
- The paper concludes that *"the scoring function used in current tree search methods still suffers from significant shortcomings"* but offers no fix or alternative.

- No ablation on the 60-tactic set: what happens with 30 or 100 tactics? How sensitive is performance?
- No ablation on pruning parameters: `α = 0.25`, `γ = 0.9`, `β` — are these optimal?
- No ablation separating data synthesis from adaptive beam size: which contributes more to the reported gains?

**Questions:**

See the weaknesses.

---

> ### Author Response · Authors · 2025-11-20
> **Response to reviewer  yJ1K (W1-W4, W5)**
>
> Thank you for the thoughtful comments. Below we provide our detailed responses to each weakness and question. For closely related questions and concerns, we provide a combined clarification.
>
> ## W1 - W4: Expression issues in the abstract, introduction, figure 1, and algorithm 1
>
> Thank you for the suggestion. In the revised submission, we have improved the phrasing and presentation in the abstract, introduction, Figure 1, and Algorithm 1. In addition, we have added an Appendix section that explains the commonly used tactics in Lean 4 to make the content easier to understand.
>
> ## W5: Comparison with whole-proof generation methods.
>
> Thank you for the question. The MiniF2F benchmark results for the 7B DeepSeek-Prover-V1.5 (whole-proof method) are shown in Table 1 below:
>
> Table 1 [1]: Performance of whole-proof generation (MiniF2F-Test)
> | Model                    | Sample budget | results      |
> |--------------------------|---------------|--------------|
> | DeepSeek-Prover-V1.5-SFT | 128           | 50.4% ± 0.4% |
> | DeepSeek-Prover-V1.5-SFT | 3200          | 53.3% ± 0.5% |
> | DeepSeek-Prover-V1.5-RL  | 128           | 51.6% ± 0.5% |
> | DeepSeek-Prover-V1.5-RL  | 3200          | 54.9% ± 0.7% |
>
> Since our evaluation uses an expansion budget of 600 (i.e., 600 model inference calls per problem), we believe this is roughly comparable to whole-proof generation methods with sample budgets of 128 or 3200, at least in terms of overall inference volume.
>
> Conceptually, whole-proof generation and tree-search represent fundamentally different paradigms, with distinct evaluation metrics and configurations (e.g., beam size, expansion limits, and global timeout in tree search). Because of these differences, a strict one-to-one comparison is difficult. Below we clarify several conceptual distincutions and our perspective on whole-proof generation:
>
> 1. **Whole-proof generation formulates theorem proving as a single-shot code-generation task.**  Given a Lean4-encoded problem, the model attempts to generate the entire proof in a single pass, which follows an end-to-end generation style. A key advantage is that evaluation is extremely efficient: unlike tree-search methods, which repeatedly query the policy model and interact with the Lean 4 prover, whole-proof generation requires only a single validity check after generation. Under pass@1, this yields very fast evaluation.
>
> 2. **However, single-shot full proof generation still faces limitations.** Because the model does not observe any intermediate proof states, it must generate a complete sequence of tactics that collectively resolve all sub-goals at once. In many nontrivial problems, each sub-goal corresponds to a logically independent part of the overall proof. The final proof often has a rough structure such as: `[snippet_1] [snippet_2] [snippet_3] ...`, where each snippet is responsible for resolving a specific sub-goal. These sub-goals do not depend on each other, yet whole-proof generation forces the model to serialize all proof steps into a single linear sequence. This mismatch makes the task difficult. This is especially challenging for smaller models (e.g., 7B), which struggle to produce long, well-structured proofs that are entirely correct in one attempt.
>
> 3. **Inference-time cost is more nuanced than it may appear.**  Tree-search methods typically use expansion budgets of a few hundred steps. To obtain a comparable number of proof candidates, whole-proof generation generally requires a large $k$ (e.g., 128 or 3200 in Table 1). We also conducted an additional experiment: using DeepSeek-Prover-V1.5, we evaluated MiniF2F-Test with pass@8 under 32 parallel processes on 4×H800 GPUs. The total runtime was roughly 1 hour. Extrapolating from pass@8 to pass@128 yields an upper-bound estimate of about 16 hours under the same setup (the actual time would be slightly faster, since problems solved early no longer require sampling). In comparison, our policy model, under the same hardware configuration, completes an equivalent workload on MiniF2F in under 2 hours using tree-search. Therefore, whole-proof generation does not necessarily offer an advantage in total inference time or computational cost when scaled to realistic sample budgets.

---

> ### Author Response · Authors · 2025-11-20
> **Response to reviewer yJ1K (W6 & W13 part 1)**
>
> ## W6 & W13: Contributions of the data synthesis method and the adaptive beam size strategy
>
> Thank you for the question. Below we further clarify the contributions of these two components.
>
> ### component 1: Data synthesis
>
> It is indeed difficult to fully quantify the contribution of our data synthesis method relative to a naive data synthesis approach, and due to time and computational constraints we were unable to run a complete set of end-to-end ablation experiments. Nevertheless, we would like to emphasize why this data synthesis procedure is necessary and why it provides meaningful value:
>
> 1. **Constrained decoding encourages diversity among generated proof steps.** During data synthesis, for a given state, the policy model tends to repeatedly generate proof steps that use the same or nearly identical tactics and premises. Even when the premises differ syntactically, they are often semantically equivalent. For example, in the MiniF2F-Test problem `imo_1997_p5`, consider the intermediate state as follow:
>
> ```
> case intro
>  nx y : ℕ
>  h : x ^ y ^ 2 = y ^ x
>  hx : 0 < x
>  hy : 0 < y
>  hne : (x, y) ≠ (1, 1)
>  ⊢ (x, y) = (1, 1)
> ```
>
> With beam size 8, BFS-Prover generates:
>
> ```
> "cases' x with x"
> "cases' lt_or_le x y with hxy hxy"
> "cases' eq_or_ne x 1 with hx1 hx1"
> "rcases eq_or_ne x 1 with (rfl | hx')"
> "cases' eq_or_ne x 1 with hx' hx'"
> "rw [Prod.mk.inj_iff] at hne ⊢"
> "cases' eq_or_ne x 1 with h' h'"
> "obtain ⟨hx', hy'⟩ := pow_pos hx 2, pow_pos hy 2"
> ```
>
> Several of these (e.g., variants of `eq_or_ne x 1` are semantically identical:
>
> ```
> "cases' eq_or_ne x 1 with hx1 hx1"
> "cases' eq_or_ne x 1 with hx' hx'"
> "cases' eq_or_ne x 1 with h' h'"
> ```
>
> In a statistical analysis on MiniF2F-Test (244 problems, B = 8, E = 600), we observed that among ~113,000 intermediate expaned nodes, about 21% contained tactic sets with pairwise similarity > 0.8 (via difflib). This behavior is often useful for solving a specific problem, but it biases the search heavily toward exploitation rather than exploration.
>
> Such repetition is undesirable for data synthesis, where the goal of data generation is to explore a wide set of reachable states (proof steps). Across problems, even when the original problems differ, sub-goals often share similar structures (e.g., many inequality proofs). Therefore, during data synthesis, encouraging the prover to diversify its generated proof steps is crucial. Constrained decoding enables this by enforcing the use of different tactics, which differ substantially in semantics and usage and thus naturally promote broader exploration.
>
> 2. **Improving coverage of common tactics.** As noted in Section 3.1, the set of commonly used tactics in Lean4 is relatively small, but each has broad applicability. For example, the `rfl` tactic can solve all *definitional equalities* in Lean4, but the model does not inherently understand its semantics. Consider the MiniF2F-Test problem `mathd_numbertheory_233`, a basic modular inverse problem: given that `b` is the multiplicative inverse of 24 modulo 121, prove that `b = 116`. Its Lean formalization is:
>
> ```
> b : ZMod (11 ^ 2)
>  h₀ : b = 24⁻¹
>  ⊢ b = 116
> ```
>
> Here, the model can easily generate the first step `norm_num[h₀]`, reducing the goal to:
>
> ```
> ⊢ 24⁻¹ = 116
> ```
>
> At this point, a single application of `rfl` could complete the proof, and Lean4 verifies it automatically. However, under a beam size of 8, the model frequently fails to generate tactic `rfl`, repeatedly proposing variants of `norm_num` instead. This causes the proof to stall, despite the simplicity of the remaining subgoal.
>
> This case highlights a key finding: when the model lacks a sufficiently broad understanding of common tactics and their usage scenarios, its overall proving ability becomes severely constrained. During data synthesis, encouraging the model to try a wider range of tactics is therefore essential. By exposing the model to diverse, tactic-rich trajectories, constrained decoding helps it learn the correct usage patterns of core tactics such as `rfl`, improving both tactic coverage and data diversity.
>
> Futhermore, we can offer an informal comparison between our data synthesis and STP. According to the STP paper, the authors performed 48 iterations of STP, generating 3.6 million conjectures, 241 million proofs, and 51.3 billion tokens in total. In contrast, our generated dataset contains approximately 20 million proof steps (i.e., proof state and corresponding proof step), roughly equivalent to 2.1 billion tokens. Since STP operates on whole proofs, 241 million proofs likely correspond to over 1 billion proof steps (assuming an average of 5 steps per proof). In terms of dataset size, ours is roughly one-fiftieth of that used to train the prover. After the model training, our one-step model achieves 60.74% and 21.18% on MiniF2F and ProofNet datasets in terms of pass@1, respectively. As a whole-proof model, STP achieves 61.2% and 19.5% on MiniF2F and ProofNet datasets in terms of pass@128, respectively.

---

> ### Author Response · Authors · 2025-11-20
> **Response to reviewer yJ1K (W6 & W13 part 2, W7)**
>
> ## W6 & W13: Contributions of the data synthesis method and the adaptive beam size strategy (cont'd)
>
> ### component 2: Adaptive beam size strategy
>
> The contribution of this strategy is reflected in Table 1 of the paper. Compared with a fixed beam size, the adaptive beam size strategy consistently improves performance across 2 benchmarks. Since this strategy does not rely on problem-specific heuristics or domain-dependent assumptions, it serves as a general-purpose enhancement to tree-search methods proof systems. Its benefits appear across different settings, demonstrating that adaptively adjusting the beam size is an effective and broadly applicable search-time optimization.
>
> ## W7: Applicability of our method to different problem types
>
> Thank you for raising this question. Here, we take the two components of our method, *data synthesis* and *adaptive beam size strategy*, to discuss what types of problems our approach is suitable for. In short, we believe that neither component exhibits any inherent bias toward specific categories of problems. Discussion details for each component are provided below:
>
> - **Data synthesis:**
>   Our data synthesis method does not inherently favor any particular type or domain of problems. Its main objective is to increase the tactic-level diversity of synthesized data. The tactics we use cover the majority of proof requirements in Lean 4 (please see W11 for further details). Therefore, given any seed dataset, our approach can be applied effectively.
>
>   If one wishes to target a specific dmoain (e.g., geometry or combinatorics), a natural extension is to incorporate additional domain-specific tactics or transformations into the commonly used tactic set. This allows the pipeline of data synthesis to adapt to specialized domains without changing the core synthesis methodology.
>
> - **Adaptive beam size:**
>   To further examine whether adaptive beam size behaves differently across problem types, we conducted an additional set of experiments using the official category labels from MiniF2F (e.g., its original competition source or mathematical subfield: *aime*, *algebra*, *amc*, *induction*, *numbertheory*, etc.) [2]. For each category, we conducted 15 independent evaluations on the MiniF2F-Test split for the fixed beam size and adaptive beam size strategies, respectively. More specifically, we conducted five runs each for fixed beam sizes B=8,16,32, and 15 runs for the adaptive beam size strategy. We then computed success rates within each category. We then computed two metrics for each category:
> * Cumulative solved: This metric measures the proportion of problems solved at least once across all runs. For example, if we evaluate 100 problems over 15 runs, and a problem is solved in any of the 15 runs, then it is counted as solved. If 82 of the 100 problems are solved at least once across all runs, then the cumulative solved score is 82/100 = 0.82.
> * Overall solved: This metric measures how often problems are solved on average across all runs. Specifically, it is the total number of solved problems across all runs divided by the product of runs×(number of problems). For example, with 100 problems and 15 runs, suppose the numbers of solved problems per run are: 70, 72, 69, 75, 74, … (15 numbers in total). If the sum across the 15 runs is 1,080 solved instances, then the overall solved metric is 1080 / (100 x 15) = 0.72.
>
> The results are shown in Table 2 and Table 3. Across all subcategories in MiniF2F-Test, the adaptive beam size strategy consistently improves performance, rather than concentrating in any particular type of problem. As a general optimization of the search process, its benefits appear broadly applicable across different problem types and mathematical domains.
>
> Table 2: Table 2 shows the performance of fixed beam size on the MiniF2F-Test split.
>
> | subcategories | cumulative solved | overall solved |
> |:---:|:---:|:---:|
> | aime |   5/15 (33.33%)   |  64/225（28.44%）  |
> | algebra |  10/18 （55.56%） |  147/270（54.44%） |
> | amc |   21/45 (46.67%)  |  245/675（36.30%） |
> | imo |   3/20 (15.00%)   |  35/300（11.67%）  |
> | induction |    5/8 (62.50%)   |  74/120（61.67%）  |
> | numbertheory |    4/8 (50.00%)   |  53/120（44.17%）  |
> | mathd_numbertheory |   51/60 (85.00%)  |  701/900（77.89%） |
> |  mathd_algebra |   60/70 (85.71%)  | 813/1050（77.43%） |
>
> Table 3: Table 3 shows the performance of adaptive beam size on the MiniF2F-Test split. The evaluation metrics are computed in the same way as in Table 2.
>
> | subcategories | cumulative solved |   overall solved   |
> |:---:|:---:|:---:|
> | aime | 5/15 (33.33%)  |  70/225（31.11%）  |
> | algebra | 10/18 （55.56%） |  149/270（55.19%） |
> | amc |   22/45 (48.89%)  |  271/675（40.15%） |
> | imo |   4/20 (20.00%)  |  46/300（15.33%）  |
> | induction | 7/8 (87.50%)   |  78/120（65.00%）  |
> | numbertheory |  4/8 (50.00%)   |  60/120（50.00%）  |
> | mathd_numbertheory |   53/60 (88.32%)  |  716/900（79.56%） |
> | mathd_algebra |  62/70 (88.57%)   | 833/1050（79.33%） |

---

> ### Author Response · Authors · 2025-11-20
> **Response to reviewer yJ1K (W8 & W10 part 1)**
>
> ## W8 & W10: Further analysis of existing scoring function
>
> Thank you for raising this point. We would like to clarify that our discussion of scoring functions is not intended to highlight a weakness of our method. Rather, our experiments reveal a broader issue shared across existing tree-search-based ATP. Most existing scoring functions are derived from the cumulative log probability of the model’s responses. In practice, this means that that the score is largely determined by perplexity: the more confident the policy model is, the higher the score will be. As a result, the effectiveness of these scoring functions depends heavily on the capability of the policy model and, consequently, on how well the model’s training distribution overlaps with the benchmark tasks.
>
> Unfortunately, most existing tree search approaches still rely on this log-probability-based scoring scheme or simple variants such as length normalization or depth normalization [3-6]. In constrast, our proposed adaptive beam size strategy acts as a passive regulator that helps mitigate some of the shortcomings of these scoring functions by adaptively constraining the search space they induce. While we do not introduce a new scoring function, the proposed adaptive beam size strategy provides a practical and effective alternative to reduce the adverse effects of relying solely on log-probability-driven search.
>
> To further illustrate the limitations of current scoring functions, we conducted an additional set of experiments and attempted a quantitative analysis to clarify where the scoring function falls short.
>
> To examine the effectiveness of different scoring functions, we selected several representative methods and conducted additional experiments on the MiniF2F-Test benchmark using our model. The experimental budget was set to K = 1, B = 8, and E = 600, with a global timeout of 1800 seconds per problem. The results are shown in Table 2 below.
>
> Table 2: Performance of representative scoring.
> | scoring function                                    | results      |
> |-----------------------------------------------------|--------------|
> | log probability                                     | 59.9% ± 0.9% |
> | log probability + depth normalization (alpha = 0.5)[1] | 59.5% ± 1.7% |
> | log prob + length normalization                 | 57.7% ± 1.7% |
> | Value network（internlm2.5-step-prover-critic [2]） | 57.2% ± 1.0% |
>
> From these results, we observe that the performance differences between scoring functions are small. Several normalization-based strategies that are theoretically appealing (such as depth normalization and length normalization) did not lead to the expected improvements. Similarly, using a value-network-based scoring function did not offer clear benefits under the current setup. For this reason, we prefer 'log probability' as the default scoring function, as it provides more stable performance across problems.

---

> ### Author Response · Authors · 2025-11-20
> **Response to reviewer yJ1K (W8 & W10 part 2, W9)**
>
> ## W8 & W10: Further analysis of existing scoring function (cont'd)
>
> To more concretely illustrate the limitations of existing scoring functions in current tree-search methods, we take the example of solving the MiniF2F-Test *mathd_algebra_598* as an example, evaluated uder the scoring function "log probability + depth normalization." The initial state of the theorem is:
>
> ```
> a b c d : ℝ
>  h₁ : 4 ^ a = 5
>  h₂ : 5 ^ b = 6
>  h₃ : 6 ^ c = 7
>  h₄ : 7 ^ d = 8
>  ⊢ a * b * c * d = 3 / 2
> ```
>
> During the search process, each state's score is computed as: *score = parent_score + current_score*, where *current_score* is obtained from the scoring function (log probability + depth normalization), and *parent_score* is the accumulated score along the path. Using this method, the search is able to correctly produce the first eight proof steps:
>
> ```
> have h₅ : (4 : ℝ) ^ a = 5 := by linarith
> have h₆ : (5 : ℝ) ^ b = 6 := h₂
> have h₇ : (6 : ℝ) ^ c = 7 := h₃
> have h₈ : (7 : ℝ) ^ d = 8 := h₄
> have h₉ : (4 : ℝ) > 0 := by norm_num
> have h₁₀ : (5 : ℝ) > 0 := by norm_num
> have h₁₁ : (6 : ℝ) > 0 := by norm_num
> have h₁₂ : (7 : ℝ) > 0 := by norm_num
> ```
>
> At the next expansion step, a correct proof step would be
> `have h₁₃ : Real.log 4 > 0 := Real.log_pos (by norm_num)`.
> However, the policy model assigns it a relatively low score (−9.29), while three other incorrect proof steps receive scores around −4. As a result, the search prefers the incorrect steps and subsequently moves into an unproductive subtree. At this point, the search depth has already reached 8. Due to depth normalization, states at different depths are mapped to a similar scale, the search struggles to escape once it enters a wrong subtree, often consuming a large portion of the budget without making progress (i.e., solve the problem). Without depth normalization, the scores of such incorrect subtrees would decrease quickly as the depth grows, allowing the search to abandon them much earlier.
>
> This example shows that depth normalization can have a negative impact in certain cases. However, this does not imply that depth normalization is always harmful. When the policy model assigns strong and consistent preferences to correct steps, depth normalization can help keep the search focused on the correct proof trajectory. Our intention with this example is to illustrate that existing scoring functions in tree-search methods can have both positive and negative effects depending on the specific problem, and each approach carries its own strengths and limitations.
>
>
> ## W9: Issues with different exploration strategies in tree search methods
>
> Thank you for your concern. We would like to clarify our conclusion drawn in Appendix A.6. First of all, we agree with your feelings that the effectiveness of the exploration strategies appear week, but this is not a weakness of our paper. Our point is that the current existing exploration strategies themselves are far less effective than commonly assumed in the field of automated theroem proving. As active researchers in this area, we conducted extensive experiments with a wide range of exploration strategies. The discussion and conclusion in Appendix A.6 are based on these trials and are intended to highlight an important observation: the exploration component in current ATP tree-search methods remains underexplored and is a valuable direction for the community to investigate further.
>
> For the details of our experiments, under the constraint of not modifying the existing scoring function, we experimented with a variety of exploration strategies. Although many of these strategies are theoretically promising, none of them led to performance improvements in practice. In fact, most of them performed worse than naïve best-first search. This outcome suggests that the underlying issue likely lies in the scoring function itself, which may have fundamental flaw when scoring the generated proof steps. We have discussed this point in detail in our responses to W8 & W10 above.

---

> ### Author Response · Authors · 2025-11-20
> **Response to reviewer yJ1K (W11)**
>
> ## W11: About the 60 commonly used tactics set
>
> Thank you for the question. We would like to clarify the criteria used to construct the set of commonly used tactics. We scanned the entire STP dataset, which contains more than 3 million samples, and computed the empirical usage frequency of all tactics. Next, we then applied a top-p filtering with a threshold of p=0.999, obtaining the 60 most frequently used tactics. These tactics cover the vast majority of proof requirements in Lean 4, particularly those related to basic mathematical structures, propositional reasoning, algebra, number theory, and analysis. Overall, they provide a comprehensive and practical foundation that balances coverage, usability, efficient learning and proving.
>
> At a high level, these tactics can be grouped into four major functional categories:
>
> - **Simplification and algebraic manipulation:** Tactics, such as `rw`, `simp`, `norm_num`, `field_simp`, `linarith`, and `ring`, are essential for symbolic computation and simplification.
>
> - **Structured reasoning:** Tactics, such as `cases`, `rcases`, and `by_cases`, support goal decomposition and case analysis for conjunctions, disjunctions, and other structured logical forms.
>
> - **Induction and constructive reasoning:** Tactics, such as `induction`, `constructor`, and `intro`, are crucial for inductive proofs, constructive arguments, and introducing hypotheses.
>
> - **Logical deduction:** Tactics, such as `contradiction`, `by_contra`, and `exfalso`, enable classical reasoning patterns such as proof by contradiction.
>
> Regarding the size of this set, we believe that reducing the number of tactics (e.g., down to ~30 tactics) would be undesirable, as several tactics in the current set play roles that cannot be easily substituted. While expanding the set is possible, most low-frequency tactics in Lean 4 can be expressed using compositions of the high-frequency tactics we include (many complex tactics are defined using simpler ones). Therefore, we prefer to allow the model to thoroughly explore and master the usage patterns of these widely applicable, commonly occurring tactics.

---

> ### Author Response · Authors · 2025-11-20
> **Response to reviewer yJ1K (W12)**
>
> ## W12: Ablation study on the pruning parameters
>
> We would like to clarify the roles of the three hyperparameters used in our data synthesis procedure:
>
> - **alpha** (depth-based exploitation-exploration control):
>   This hyperparameter controls the balance between exploitation and exploration with respect to tree depth in the search tree. Given a beam size of $B$, we retain only $\alpha B$ branches at each step. During data synthesis, we typically set $B = 32$ to allow the policy model to generate tactics to which it assigns relatively high confidence. However, such a large beam size makes the search tree excessively wide, making it difficult to obtain new states that differ significantly from the original state after several proof steps. This effect is also visible during evaluation (e.g., on MiniF2F, where the policy model is relatively familiar with the domain, a beam size of 32 is not optimal.). By keeping only $\alpha B$ branches, we obtain more nodes from deeper parts of the tree, improving exploration.
>
> - **beta** (branch-wise exploitation-exploration control):
>   This hyperparameter regulates the trade-off across different branches within beam search. As discussed in part 1, naïve data synthesis tends to overemphasize exploitation, while pure constrained decoding risks pushing the model into excessively exploratory behavior, forgetting solving the problem. To avoid both extremes, we include the top-$\beta$ most likely branches (ensuring exploitation) and randomly sample additional branches from the remaining candidates (ensuring exploration). This mixture maintains diversity in the synthesized data while preventing the policy model from drifting too far from problem-solving behavior.
>
> - **gamma** (budget decay for easy problems):
>   This hyperparameter helps terminate the search early for problems that are too easy. In the STP dataset, some problems require only a few steps to reach a proof-finished state. Unlike evaluation (where search terminates as soon as one proof is found), data synthesis aims to discover multiple distinct proof-finished paths. However, allocating the same search budget to every seed problem is not appropriate. Therefore, each time a new proof-finished path is discovered, the remaining search budget is multiplied by a decay factor $\gamma < 1$. Easy problems naturally yield multiple finished paths, causing their budgets to decay quickly and terminating the search earlier. A smaller $\gamma < 1$ results in faster decay.
>
> In practice, our data synthesis pipeline is designed for million-scale sample generation, and data is produced in multiple batches. For each batch, we make small adjustments to the pruning hyperparameters based on the quality of the previous batch. The recommended pruning hyperparameters reported in the paper are the result of this iterative refinement process. Because of this data synthesis workflow, there is no single global hyperparameter configuration that would meaningfully support a fully exhaustive ablation. Given constraints on time and computational resources, it is unfortunately not feasible for us to rerun multiple full-scale synthesis processes to produce complete ablations.
>
> ---
>
> Reference:
>
> [1] Xin, H., Ren, Z. Z., Song, J., Shao, Z., Zhao, W., Wang, H., ... & Ruan, C. (2024). Deepseek-prover-v1. 5: Harnessing proof assistant feedback for reinforcement learning and monte-carlo tree search. arXiv preprint arXiv:2408.08152.
>
> [2] Zheng, K., Han, J. M., & Polu, S. (2021). Minif2f: a cross-system benchmark for formal olympiad-level mathematics. arXiv preprint arXiv:2109.00110.
>
> [3] Wu, Z., Huang, S., Zhou, Z., Ying, H., Wang, J., Lin, D., & Chen, K. (2024). Internlm2. 5-stepprover: Advancing automated theorem proving via expert iteration on large-scale lean problems. arXiv preprint arXiv:2410.15700.
>
> [4] Xin, R., Xi, C., Yang, J., Chen, F., Wu, H., Xiao, X., ... & Ding, M. (2025, July). Bfs-prover: Scalable best-first tree search for llm-based automatic theorem proving. In Proceedings of the 63rd Annual Meeting of the Association for Computational Linguistics (Volume 1: Long Papers) (pp. 32588-32599).
>
> [5] Liang, Z., Song, L., Li, Y., Yang, T., Zhang, F., Mi, H., & Yu, D. (2025). MPS-Prover: Advancing Stepwise Theorem Proving by Multi-Perspective Search and Data Curation. arXiv preprint arXiv:2505.10962.
>
> [6] Shen, Z., Huang, N., Yang, F., Wang, Y., Gao, G., Xu, T., ... & Dong, B. (2025). REAL-Prover: Retrieval Augmented Lean Prover for Mathematical Reasoning. arXiv preprint arXiv:2505.20613.

---

### Official Review · Reviewer_wuPN · 2025-10-31

**Soundness:** 3
**Presentation:** 4
**Contribution:** 3
**Rating:** 6
**Confidence:** 4

**Summary:**

This paper presents a proof-state exploration framework for automated theorem proving in Lean 4, aiming to improve LLM-based provers through self-supervised data generation.

Starting from existing formal theorems (mainly Mathlib and STP), the authors perform beam-based exploration of proof states guided by a policy model (Qwen2.5-Math-7B). They propose two key components: proof-state exploration with heuristic pruning and adaptive beam-size search.

The resulting synthetic dataset is used to fine-tune the base model, yielding a policy that achieves 60.74 % Pass@1 on MiniF2F and 21.18 % Pass@1 on ProofNet, surpassing several prior Lean-based provers.

**Strengths:**

1. The paper formulates a well-structured pipeline for data synthesis via proof-state exploration, bridging self-improvement and verifiable proof checking in Lean 4.
2. The reported 60.74 % Pass@1 on MiniF2F is competitive with or better than recent specialized theorem-proving LLMs, suggesting tangible benefits from the proposed training pipeline.
3. The proposed adaptive beam-size strategy which dynamically shrinking the beam with depth is a simple yet effective heuristic that mitigates local-trap issues in tree search and improves computational efficiency without sacrificing success rate.
4. The inclusion of a BLEU-based decontamination step against MiniF2F and ProofNet benchmarks is a necessary step that shows the authors’ awareness of data leakage concerns.

**Weaknesses:**

My main concern is that the paper does not report the raw (zero-shot) performance of Qwen2.5-Math-7B on MiniF2F or ProofNet. As a result, it is unclear how much improvement stems from the exploration-based fine-tuning versus the inherent strength of the base model. For fair comparison, the authors should either (a) report base model results, or (b) fine-tune the same base models used by prior baselines (e.g., BFSProver) under the same training pipeline.

**Questions:**

1. Could the authors report the zero-shot performance of the base Qwen2.5-Math-7B model on MiniF2F and ProofNet? This would clarify the absolute contribution of the exploration-based fine-tuning.
2. Did you evaluate how many generated samples were removed during BLEU-based decontamination?
3. Is the adaptive beam-size heuristic compatible with parallel expansion or batched inference during search, or is it inherently sequential?

---

> ### Author Response · Authors · 2025-11-20
> **Response to reviewer wuPN (W1 & Q1, Q2)**
>
> Thank you for the thoughtful comments. Below we provide our detailed responses to each weakness and question. For closely related questions and concerns, we provide a combined clarification.
>
> ## W1 & Q1: Performance of Qwen2.5-Math-7B on MiniF2F and ProofNet.
>
> Thank you for raising this point. We conducted additional experiments using the Qwen2.5-Math-7B base model under the same experimental configuration as in our paper. The results are summarized in Table 1 below.
>
> Table 1. Performance of Qwen2.5-Math-7B.
> | Dataset  | Prompt Type                   | Avg pass@2 |
> |----------|-------------------------------|:----------:|
> | MiniF2F  | zero-shot（prompt1）          |    4.71%   |
> |          | few-shot （prompt2）          |    8.20%   |
> |          | zero-shot-tactics （prompt3） |    3.69%   |
> |          | few-shot-tactics （prompt4）  |    9.02%   |
> | ProofNet | zero-shot  (prompt1)          |    0.00%   |
> |          | few-shot （prompt2）          |    0.82%   |
> |          | zero-shot-tactics （prompt3） |    0.00%   |
> |          | few-shot-tactics （prompt4）  |    0.00%   |
>
> Under the same experimental configuration, the performance of the Qwen2.5-Math-7B base model is far from satisfactory. Approximately 35% of the problems fail to progress after the first expansion, as all generated tactics fail to execute. Across all problems, the maximum number of successful expansions is only three. Problems that are solved are almost exclusively trivial ones requiring a single step with basic tactics such as `ring`, `linarith`, `norm_num`, or `simp`. This level of performance is insufficient for supporting a tree-search-based Lean 4 automated theorem proving pipeline.
>
> Based on our observations, the main limitations of the Qwen2.5-Math-7B base model fall into two main categories:
>
> 1. **Unreliable output formatting.** The current automated proving system relies on extracting executable Lean 4 code snippets from the model output. However, the 7B base model struggles to reliably follow the required output format, even under a wide range of prompting strategies (including zero-shot, one-shot, few-shot, and explicitly listing common tactics in the prompt). Despite these efforts, correctly extracting Lean 4 code also remains challenging. Our statistics show that only about 20% of the base model's responses meet the format requirements and can be successfully parsed into Lean 4 code snippets.
>
> 2. **Low execution success rate of generated proof steps.** Among the extracted proof steps, only a small fraction can be executed successfully by Lean 4 server. The model demonstrates limited understanding of Lean 4 syntax and semantics, often producing steps with low tactical diversity, missing premises information, or syntax errors, making them largely ineffective in advancing the proof process.
>
> Overall, at its current model size and configuration, the Qwen2.5-Math-7B base model lacks sufficient capability to reliably support tree-search-methods-based Lean 4 automated theorem proving.
>
> ## Q2: Samples filtered during the decontamination process.
>
> Thank you for the question. We performed decontamination using the BLEU score implementation from the NLTK library, applying a threshold of 0.6. This procedure removed approximately 90k samples, accounting for about 0.02% of the full dataset.
>
> More specifically, during filtering, we computed the BLEU score between every state encountered during exploration and every problem in the benchmark. For each state, we took the maximum BLEU score over all benchmark problems (i.e., the problems in MiniF2F test set). If this maximum score exceeded the threshold, we considered the state overly similar to at least one benchmark problem and removed the corresponding sample.
>
> To better understand the BLEU score distribution, we partitioned the BLEU score range $[0, 1]$ into ten uniformly sized intervals. The proportion of samples falling into each interval is shown in Table 2 below:
>
> Table 2. BLEU score distribution among synthesized samples.
> | score range | proportion |
> |-------------|------------|
> | [0.0, 0,1)  | 22.20%     |
> | [0.1, 0.2)  | 41.13%     |
> | [0.2, 0.3)  | 27.94%     |
> | [0.3, 0.4)  | 7.16%      |
> | [0.4, 0.5)  | 1.38%      |
> | [0.5, 0.6)  | 0.17%      |
> | [0.6, 0.7)  | 0.02%      |
> | [0.7, 0.8)  | <0.01%     |
> | [0.8, 0.9)  | <0.01%     |
> | [0.9, 1.0]  | <0.01%     |
>
> Based on manual inspection, samples with scores in the range $[0.5, 0.6)$ often share similar expressions with benchmark problems, but the underlying problems remain different. In contrast, scores above 0.6 typically include that the underlying problems themselves are noticeably similar. This empirical observation motivated our choice of **0.6** as the final threshold.

---

> ### Author Response · Authors · 2025-11-20
> **Response to reviewer wuPN (Q3)**
>
> ## Q3: Compatibility of the adaptive beam size strategy with other methods during the search process
>
> Thank you for raising this question. Once the beam size is determined for each state to be expanded, the model’s inference process is identical to that of standard beam search. Therefore, our adaptive beam size strategy is fully compatible with both parallel expansion and batched inference methods.
>
> In practical evaluations, as we explain in Appendix A.5, different problems produce search trees that vary greatly in both size and search speed. To manage this efficiently, we launch dozens of tree search workers in parallel, each paired with one of several policy model instances. Each tree search worker sends its current search state to a model instance for inference. The policy model is implemented as an AsyncLLMEngine instance in vLLM, which allows it to asynchronously serve requests from multiple tree search workers without blocking.

---

### Official Review · Reviewer_GV4S · 2025-11-04

**Soundness:** 3
**Presentation:** 3
**Contribution:** 3
**Rating:** 6
**Confidence:** 3

**Summary:**

This paper proposes a scalable data synthesis framework for LLM-based automated theorem proving. The core idea, Proof State Exploration performs constrained decoding over a curated tactic set to systematically explore intermediate proof states, generating diverse (state, tactic, next state) triples for SFT without relying on inefficient expert iteration. In addition, an adaptive beam size strategy dynamically adjusts search breadth during proof generation, balancing exploration and exploitation.

**Strengths:**

1. The proposed proof state exploration pipeline provides a well-structured and scalable solution for synthetic data generation and the idea of decoupling tactic exploration from premise generation and forcing low-probability tactic sampling is well motivated
2. The experimental results are sound, with detailed comparisons, ablation on beam size, and transparent computational settings

**Weaknesses:**

While the method is conceptually strong, it remains largely heuristic.
1. The pruning and beam-decay hyperparameters are fixed ad hoc without sensitivity analysis or further ablations
2. The methodology section is long and very formalized, but lacks intuition for key ideas like why constrained decoding enhances tactic diversity or how pruning parameters (α, β, γ) are chosen and affect synthesis efficiency
3. The scoring function used in tree search still relies on local log-probabilities in Eq. 1, which the paper itself admits to be suboptimal but no quantitative evidence is given for failure case analysis
4. the exploration may remain bounded to “local neighborhoods” of seed datasets, raising concerns about out-of-distribution generalization, especially beyond ProofNet

**Questions:**

1. Could the adaptive beam control learned instead of manually scheduled, like via a small controller model?
2. I think ProofNet differs significantly from Lean distributions, how does the model handle domain shift, was any domain-specific adaptation adopted?

---

> ### Author Response · Authors · 2025-11-20
> **Response to reviewer GV4S (W1)**
>
> Thank you for the thoughtful comments. Below we provide our detailed responses to each weakness and question. For closely related questions and concerns, we provide a combined clarification.
>
> ## W1: Further analysis of pruning and beam-decay hyperparameters
>
> Thank you for your question. Here we provide additional discussion of the two hyperparameters:
>
> 1. **Pruning hyperparameters.** As described in our response to W2 part 2, the intuition for the three pruning hyperparameters is already discussed in detail. In practice, our data synthesis pipeline is designed for million-scale sample generation, and data is produced in multiple batches. For each batch, we make small adjustments to the pruning hyperparameters based on the quality of the previous batch. The recommended pruning hyperparameters reported in the paper are the result of this iterative refinement process. Because of this data synthesis workflow, there is no single global hyperparameter configuration that would meaningfully support a fully exhaustive ablation. Given constraints on time and computational resources, it is unfortunately not feasible for us to rerun multiple full-scale synthesis processes to produce complete ablations.
>
> 2. **Beam-decay hyperparameters.** For the adaptive beam size strategy, we conducted an additional set of experiments on MiniF2F using our model as the policy model. We varied the three hyperparameters and evaluated each configuration twice. The results are shown in Table 1. The results indicate that as long as $B_{\max}$ and $B_{\min}$ fall within a reasonable range, the exact choices of the three hyperparameters have limited influence on evaluation performance.
> - For $B_{\max}$: Given a fixed benchmark, let $B^\*$ denote the optimal beam size under fixed-beam search. We recommend selecting $B_{\max}$ slightly larger than $B^\*$, but not exceeding $2B^\*$.
> - For $B_{\min}$: We recommend values of 2 or 4, and empirically we did not observe a significant difference between these values. Since $B_{\min}$ controls the beam size in deeper layers of the search tree, using a relatively small value slightly improves pass@1.
>
> Table 1. Effect of varying beam-decay hyperparameters on MiniF2F.
> | $B_{\max}$ | $B_{\min}$ | $\lambda$ | Avg pass@1 |
> |-------|-------|--------|------------|
> | 16    | 4     | 15     | 60.74%     |
> | 16    | 4     | 5      | 60.45%     |
> | 16    | 4     | 30     | 60.04%     |
> |       |       |        |            |
> | 32    | 16    | 15     | 59.22%     |
> | 32    | 4     | 15     | 60.04%     |
> | 16    | 2     | 15     | 60.04%     |
> | 8     | 2     | 15     | 59.22%     |

---

> ### Author Response · Authors · 2025-11-20
> **Response to reviewer GV4S (W2 part1)**
>
> ## W2 part 1: Intuition behind using constrained decoding to improve data diversity
>
> Thank you for the question. We would like to clarify the intuition behind using constrained decoding during data synthesis process:
>
> 1. **Constrained decoding encourages diversity among generated proof steps.** During data synthesis, for a given state, the policy model tends to repeatedly generate proof steps that use the same or nearly identical tactics and premises. Even when the premises differ syntactically, they are often semantically equivalent. For example, in the MiniF2F-Test problem `imo_1997_p5`, consider the intermediate state as follow:
>
> ```
> case intro
>  nx y : ℕ
>  h : x ^ y ^ 2 = y ^ x
>  hx : 0 < x
>  hy : 0 < y
>  hne : (x, y) ≠ (1, 1)
>  ⊢ (x, y) = (1, 1)
> ```
>
> With beam size 8, BFS-Prover generates tactics including:
>
> ```
> "cases' x with x"
> "cases' lt_or_le x y with hxy hxy"
> "cases' eq_or_ne x 1 with hx1 hx1"
> "rcases eq_or_ne x 1 with (rfl | hx')"
> "cases' eq_or_ne x 1 with hx' hx'"
> "rw [Prod.mk.inj_iff] at hne ⊢"
> "cases' eq_or_ne x 1 with h' h'"
> "obtain ⟨hx', hy'⟩ := pow_pos hx 2, pow_pos hy 2"
> ```
>
> Several of these (e.g., variants of `eq_or_ne x 1` are semantically identical:
>
> ```
> "cases' eq_or_ne x 1 with hx1 hx1"
> "cases' eq_or_ne x 1 with hx' hx'"
> "cases' eq_or_ne x 1 with h' h'"
> ```
>
> In a small statistical analysis on MiniF2F-Test (244 problems, beam size = 8, num_expansions = 600, timeout = 1800), we observed that among ~113,000 intermediate expaned nodes, about 21% contained tactic sets with pairwise similarity > 0.8 (via `difflib.SequenceMatcher`). This behavior is often useful for solving a specific problem, but it biases the search heavily toward exploitation rather than exploration.
>
> Such repetition is undesirable for data synthesis, where the goal of data generation is to explore a wide set of reachable states (proof steps). Across problems, even when the original problems differ, sub-goals often share similar structures (e.g., many inequality proofs). Therefore, during data synthesis, encouraging the prover to diversify its generated proof steps is crucial. Constrained decoding enables this by enforcing the use of different tactics, which differ substantially in semantics and usage and thus naturally promote broader exploration.
>
> 2. **Improving coverage of common tactics.** As noted in Section 3.1, the set of commonly used tactics in Lean4 is relatively small, but each has broad applicability. For example, the `rfl` tactic can solve all *definitional equalities* in Lean4, but the model does not inherently understand its semantics. Consider the MiniF2F-Test problem `mathd_numbertheory_233`, a basic modular inverse problem: given that `b` is the multiplicative inverse of 24 modulo 121, prove that `b = 116`. Its Lean formalization is:
>
> ```
> b : ZMod (11 ^ 2)
>  h₀ : b = 24⁻¹
>  ⊢ b = 116
> ```
>
> Here, the model can easily generate the first step `norm_num[h₀]`, reducing the goal to:
>
> ```
> ⊢ 24⁻¹ = 116
> ```
>
> At this point, a single application of `rfl` could complete the proof, and Lean4 verifies it automatically. However, under a beam size of 8, the model frequently fails to generate tactic `rfl`, repeatedly proposing variants of `norm_num` instead. This causes the proof to stall, despite the simplicity of the remaining subgoal.
>
> This case highlights a key finding: when the model lacks a sufficiently broad understanding of common tactics and their usage scenarios, its overall proving ability becomes severely constrained. During data synthesis, encouraging the model to try a wider range of tactics is therefore essential. By exposing the model to diverse, tactic-rich trajectories, constrained decoding helps it learn the correct usage patterns of core tactics such as `rfl`, improving both tactic coverage and data diversity.
>
> Furthermore, we can offer an informal comparison between our data synthesis and STP. According to the STP paper, the authors performed 48 iterations of STP, generating 3.6 million conjectures, 241 million proofs, and 51.3 billion tokens in total. In contrast, our generated dataset contains approximately 20 million proof steps (i.e., proof state and corresponding proof step), roughly equivalent to 2.1 billion tokens. Since STP operates on whole proofs, 241 million proofs likely correspond to over 1 billion proof steps (assuming an average of 5 steps per proof). In terms of dataset size, ours is roughly one-fiftieth of that used to train the prover. After the model training, our one-step model achieves 60.74% and 21.18% on MiniF2F-test and ProofNet-test datasets in terms of pass@1, respectively. As a whole-proof model, STP achieves 61.2% and 19.5% on MiniF2F-test and ProofNet-test datasets in terms of pass@128, respectively.

---

> ### Author Response · Authors · 2025-11-20
> **Response to reviewer GV4S (W2 part2)**
>
> ## W2 part 2: Hyperparameter choices in the data synthesis process
>
> We would like to clarify the roles of the three hyperparameters used in our data synthesis procedure:
>
> - **alpha** (depth-based exploitation-exploration control):
>   This hyperparameter controls the balance between exploitation and exploration with respect to tree depth in the search tree. Given a beam size of $B$, we retain only $\alpha B$ branches at each step. During data synthesis, we typically set $B = 32$ to allow the policy model to generate tactics to which it assigns relatively high confidence. However, such a large beam size makes the search tree excessively wide, making it difficult to obtain new states that differ significantly from the original state after several proof steps. This effect is also visible during evaluation (e.g., on MiniF2F, where the policy model is relatively familiar with the domain, a beam size of 32 is not optimal.). By keeping only $\alpha B$ branches, we obtain more nodes from deeper parts of the tree, improving exploration.
>
> - **beta** (branch-wise exploitation-exploration control):
>   This hyperparameter regulates the trade-off across different branches within beam search. As discussed in part 1, naïve data synthesis tends to overemphasize exploitation, while pure constrained decoding risks pushing the model into excessively exploratory behavior, forgetting solving the problem. To avoid both extremes, we include the top-$\beta$ most likely branches (ensuring exploitation) and randomly sample additional branches from the remaining candidates (ensuring exploration). This mixture maintains diversity in the synthesized data while preventing the policy model from drifting too far from problem-solving behavior.
>
> - **gamma** (budget decay for easy problems):
>   This hyperparameter helps terminate the search early for problems that are too easy. In the STP dataset, some problems require only a few steps to reach a proof-finished state. Unlike evaluation (where search terminates as soon as one proof is found), data synthesis aims to discover multiple distinct proof-finished paths. However, allocating the same search budget to every seed problem is not appropriate. Therefore, each time a new proof-finished path is discovered, the remaining search budget is multiplied by a decay factor $\gamma < 1$. Easy problems naturally yield multiple finished paths, causing their budgets to decay quickly and terminating the search earlier. A smaller $\gamma < 1$ results in faster decay.

---

> ### Author Response · Authors · 2025-11-20
> **Response to reviewer GV4S (W3)**
>
> ## W3: Further analysis of issues in existing scoring functions
>
> Thank you for raising this point. To examine the effectiveness of different scoring functions, we selected several representative methods and conducted additional experiments on the MiniF2F-Test benchmark using our model. The experimental budget was set to K = 1, B = 8, and E = 600, with a global timeout of 1800 seconds per problem. The results are shown in Table 2 below.
>
> Table 2: Performance of representative scoring.
> | scoring function                                    | results      |
> |-----------------------------------------------------|--------------|
> | log probability                                     | 59.9% ± 0.9% |
> | log probability + depth normalization (alpha = 0.5) [1] | 59.5% ± 1.7% |
> | log prob + length normalization                 | 57.7% ± 1.7% |
> | Value network（internlm2.5-step-prover-critic [2]） | 57.2% ± 1.0% |
>
> From these results, we observe that the performance differences between scoring functions are small. Several normalization-based strategies that are theoretically appealing (such as depth normalization and length normalization) did not lead to the expected improvements. Similarly, using a value-network-based scoring function did not offer clear benefits under the current setup. For this reason, we prefer 'log probability' as the default scoring function, as it provides more stable performance across problems.
>
> To more concretely illustrate the limitations of existing scoring functions in current tree-search methods, we take the example of solving the MiniF2F-Test *mathd_algebra_598* as an example, evaluated uder the scoring function "log probability + depth normalization." The initial state of the theorem is:
>
> ```
> a b c d : ℝ
>  h₁ : 4 ^ a = 5
>  h₂ : 5 ^ b = 6
>  h₃ : 6 ^ c = 7
>  h₄ : 7 ^ d = 8
>  ⊢ a * b * c * d = 3 / 2
> ```
>
> During the search process, each state's score is computed as: *score = parent_score + current_score*, where *current_score* is obtained from the scoring function (log probability + depth normalization), and *parent_score* is the accumulated score along the path. Using this method, the search is able to correctly produce the first eight proof steps:
>
> ```
> have h₅ : (4 : ℝ) ^ a = 5 := by linarith
> have h₆ : (5 : ℝ) ^ b = 6 := h₂
> have h₇ : (6 : ℝ) ^ c = 7 := h₃
> have h₈ : (7 : ℝ) ^ d = 8 := h₄
> have h₉ : (4 : ℝ) > 0 := by norm_num
> have h₁₀ : (5 : ℝ) > 0 := by norm_num
> have h₁₁ : (6 : ℝ) > 0 := by norm_num
> have h₁₂ : (7 : ℝ) > 0 := by norm_num
> ```
>
> At the next expansion step, a correct proof step would be
> `have h₁₃ : Real.log 4 > 0 := Real.log_pos (by norm_num)`.
> However, the policy model assigns it a relatively low score (−9.29), while three other incorrect proof steps receive scores around −4. As a result, the search prefers the incorrect steps and subsequently moves into an unproductive subtree. At this point, the search depth has already reached 8. Due to depth normalization, states at different depths are mapped to a similar scale, the search struggles to escape once it enters a wrong subtree, often consuming a large portion of the budget without making progress (i.e., solve the problem). Without depth normalization, the scores of such incorrect subtrees would decrease quickly as the depth grows, allowing the search to abandon them much earlier.
>
> This example shows that depth normalization can have a negative impact in certain cases. However, this does not imply that depth normalization is always harmful. When the policy model assigns strong and consistent preferences to correct steps, depth normalization can help keep the search focused on the correct proof trajectory. Our intention with this example is to illustrate that existing scoring functions in tree-search methods can have both positive and negative effects depending on the specific problem, and each approach carries its own strengths and limitations.

---

> ### Author Response · Authors · 2025-11-20
> **Response to reviewer GV4S (W4 & Q2, Q1)**
>
> ## W4 & Q2: Generalization risks on ProofNet and handling domain shift
>
> Thank you for your question. We agree that ProofNet differs substantially from our synthesized Lean4 corpus in data distribution. We did not apply any domain-specific adaptation for this benchmark. As noted in W2, our data synthesis approach relies on the assumption that, within the same domain, different original problems often share highly similar sub-goals during the reasoning process. This assumption is reasonable for Lean-style mathematical proofs, which limits generalization to domains with different distributions. We believe that other data-synthesis pipeline built on a single-domain seed dataset would face the same limitation: the resulting policy model does not automatically acquire robustness to large domain shifts.
>
> Although the policy model cannot directly handle this shift, we can partially compensate for the gap by increasing the beam size during evaluation. On ProofNet,    where the training data contain relatively few samples from related distributions, the model generates far fewer high-quality branches during beam search compared to MiniF2F. A larger beam size increases the number of candidate branches, thereby improving the chance of finding a valid proof path. This behavior is consistent with the evaluation trends reported in Table 1 of the paper.
>
> ## Q1: Can the beam size be controlled by a small learned model?
>
> Thank you for the thoughtful question. Indeed, after recognizing how crucial the beam size is in the tree search process, we initially explored the idea of training a controller model that predicts an appropriate beam size for each state. Our hope was that such a controller might partially replace the role of a value network. However, we found this approach challenging in practice for several reasons:
>
> 1. **Beam size is typically treated as a manual hyperparameter in LLM decoding.** Beam search is one of widely used decoding strategies in the LLM community, where beam size is almost universally treated as a decoding hyperparameter, analogous to *top-k* or *top-p*, and is manually set to shape model behavior, rather than predicted by an auxiliary model. For this reason, we view our adaptive beam size strategy as complementary to, rather than a replacement for, the value network: the adaptive beam size prunes the search tree, while the value network can further prioritize promising branches.
>
> 2. **Training such a controller efficiently is extremely difficult.** A supervised approach is infeasible, as it is nearly impossible to collect labeled data specifying the “optimal” beam size for each state. An RL-based approach is theoretically possible but operationally prohibitive. The controller would need to select beam sizes over a discrete set, and training would require maintaining multiple tree structures within a single batch. These trees progress at different speeds, leading to substantial system complexity and computational cost.
>
> 3. **Strong coupling between the controller and the policy model.** Such a controller would be tightly coupled with the specific policy model used in the tree search, suffering from the same issues as a value network. As discussed in Section 5, the optimal beam size heavily depends on the policy model and its training data. For example, BFS-Prover, trained via expert iteration only on successful proof paths, tends to pursue a single promising proof trajectory more deeply, making smaller beam sizes effective. In contrast, our synthesized dataset emphasizes diversity, requiring a larger beam size to fully exploit the model’s capabilities.
>
> 4. **Mismatch between controller input and the model-dependent nature of beam size.** During tree search, the controller must decide the beam size *before* the policy model generates candidates. Thus, it only observes the current state and has no access to any policy-model specific information. This contradicts our empirical observation that optimal beam size is strongly model-dependent.
>
> Given these challenges, we temporarily decided not to pursue this direction further for now. That said, we agree that learning to control beam size is an intriguing idea, and we are exploring alternative designs that may avoid the issues outlined above.
>
> ---
>
> Reference:
>
> [1] Xin, R., Xi, C., Yang, J., Chen, F., Wu, H., Xiao, X., ... & Ding, M. (2025, July). Bfs-prover: Scalable best-first tree search for llm-based automatic theorem proving. In Proceedings of the 63rd Annual Meeting of the Association for Computational Linguistics (Volume 1: Long Papers) (pp. 32588-32599).
>
> [2] Wu, Z., Huang, S., Zhou, Z., Ying, H., Wang, J., Lin, D., & Chen, K. (2024). Internlm2. 5-stepprover: Advancing automated theorem proving via expert iteration on large-scale lean problems. arXiv preprint arXiv:2410.15700.

---

### Official Review · Reviewer_oTgp · 2025-11-18

**Soundness:** 2
**Presentation:** 3
**Contribution:** 2
**Rating:** 4
**Confidence:** 3

**Summary:**

The paper proposes a scalable synthetic–data generation pipeline for Lean 4 theorem proving by exploring intermediate proof states using constrained decoding, premise completion, and validation through Lean execution, producing roughly 20M high-quality proof transitions for supervised fine-tuning. Combined with an adaptive beam-decay strategy for inference, this approach yields substantial performance gains over strong baselines such as InternLM2.5-StepProver and BFS-Prover on MiniF2F and ProofNet, showing that large-scale, systematically generated synthetic data is key to improving LLM-based automated theorem proving.

**Strengths:**

- Well written and easy to follow; the illustrative figures and sampled examples are particularly helpful.
- Code is easy to access, and the experiments are reproducible within an academic compute budget.

**Weaknesses:**

- As a data synthesis strategy, I don’t think it has been critically evaluated: How well does the synthetic data perform under standard search techniques (e.g., greedy decoding)? How much does the exploration component actually contribute to the final proof success rate? An ablation study would help clarify this.
- It would be helpful to include an algorithmic comparison between the newly proposed tree-search method and existing approaches, rather than only reporting raw performance. It would also be valuable to see how the main baseline, BFS-Prover, performs with a slightly larger compute budget.

**Questions:**

- In Table 1, should I assume that your models are fine-tuned on the synthetic dataset described in Section 3.1, while the baselines InternLM2.5-StepProver and BFS-Prover are not?

---

> ### Author Response · Authors · 2025-11-20
> **Response to reviewer oTgp (W1)**
>
> Thank you for the thoughtful comments. Below we provide our detailed responses to each weakness and question. For closely related questions and concerns, we provide a combined clarification.
>
> ## W1: Concern regarding the data synthesis strategy
>
> Thank you for raising this question. We fully agree with your concern, and we appreciate your constructive suggestion regarding an ablation study on data synthesis strategy. Following your advice, we conducted additional ablation experiments in two days, and we report the corresponding results below.
>
> Specifically, we used the same policy model as in the paper and performed data synthesis on the same seed dataset (STP) using a greedy decoding strategy. Starting from approximately 60,000 randomly sampled problems as seeds, we obtained about 1.04 million state–tactic pairs as the raw synthesized data. We then removed samples that produced errors when validated by the Lean prover and applied the same data contamination checks as in the main paper. After filtering, roughly 580,000 samples remained, forming the final greedy-decoding-based synthesized dataset used for training.
>
> To ensure a fair comparison, we randomly sampled an equal-sized subset of 580,000 examples from our own synthesized dataset. Both datasets were used to fine-tune the same base model, Qwen2.5-Math-7B, and were evaluated on MiniF2F-Test with identical budgets and experimental settings. The results are shown in Table 1 below.
>
> Table 1: Comparison of data synthesis strategies.
> |                                                         | greedy decoding | our data synthesis method |
> |:-------------------------------------------------------:|:---------------:|:-------------------------:|
> |                       1 x 8 x 600                       |      51.64%     |           55.74%          |
> |                       1 x 16 x 600                      |      50.82%     |           56.15%          |
> |                       1 x 32 x 600                      |      50.00%     |           54.92%          |
> | adaptive beam size stratedy (setting same as our paper) |      52.87%     |           56.15%          |
>
> These results show that our data synthesis method can improve the diversity of generated training examples. This diversity helps the model better understand the applicable contexts of different tactics and enables it to explore a broader set of solution strategies during problem solving. In contrast, data synthesized with greedy decoding tends to overemphasize exploitation. This issue is especially pronounced when the policy model has been trained through expert iteration, as the model often produces highly similar proof steps when encountering the same state. When such homogeneous data is used for training, the resulting policy model tends to generate proof steps that are very similar to one another, which limits overall performance.

---

> ### Author Response · Authors · 2025-11-20
> **Response to reviewer oTgp (W2, Q1)**
>
> ## W2: Algorithmic comparison and larger budget evaluation
>
> Thank you for your suggestion. Below we provide a clarification of how our adaptive beam size strategy compares to other search strategies at the methodological level.
>
> In tree-search methods, the high-level workflow is largely consistent across different approaches: the search proceeds on a tree structure, repeatedly selecting one node to expand until a valid proof is found. Once the policy model is fixed, the major differences between existing search strategies primarily arise from how they design their scoring functions. Common examples of score function include depth normalization, length normalization, or other heuristic scoring terms. These scoring functions are generally aim to balance how different proof steps (i.e., the steps inner one proof state and the steps inter different proof states) are selected. For instance, depth normalization increases the scores of deeper nodes to make them more likely to be choosen by the tree-search, and length normalization adjusts the scores based on the length of the generated proof step so that longer steps have a higher chance of being chosen.
>
> However, models trained with our data synthesis strategy behave differently when directly combined with these existing scoring schemes. Because our synthesized training data encourages exploration rather than exploitation, the resulting policy model does not naturally commit to a single proof path. In practice, under the same expansion budget, our policy model tends to generate a search tree that is wider and shallower. As a result, an additional mechanism is required to guide the model toward more focused search paths. This is exactly the role of our adaptive beam size strategy, which can be viewed as a pruning mechanism. It determines whether the search should favor exploration or exploitation based on both the depth of the current node and the remaining search budget. As the search proceeds, this mechanism gradually helps the policy model narrow down its search and produce more concentrated proof step candidates as the search progresses.
>
> Regarding the baseline evaluation with a larger budget, due to limited time, we conduct five runs of BFS-Prover on MiniF2F-Test using a budget of K = 1, B = 16, and E = 600. The result is 58.40% ± 0.56%.
>
> ## Q1: The training data used for our model and the baseline
>
> Thank you for your question. Your understanding is correct. The training datasets for both InternLM2.5-StepProver and BFS-Prover are constructed using the Expert Iteration [1,2]. In Expert Iteration, multiple rounds of search are performed on a seed dataset to obtain proof steps from the valid trajectories, and then trains the model iteratively.
>
> ---
>
> Reference:
>
> [1] Wu, Z., Huang, S., Zhou, Z., Ying, H., Wang, J., Lin, D., & Chen, K. (2024). Internlm2. 5-stepprover: Advancing automated theorem proving via expert iteration on large-scale lean problems. arXiv preprint arXiv:2410.15700.
>
> [2] Xin, R., Xi, C., Yang, J., Chen, F., Wu, H., Xiao, X., ... & Ding, M. (2025, July). Bfs-prover: Scalable best-first tree search for llm-based automatic theorem proving. In Proceedings of the 63rd Annual Meeting of the Association for Computational Linguistics (Volume 1: Long Papers) (pp. 32588-32599).

---

### Author Response · Authors · 2025-11-26

We sincerely appreciate the valuable suggestions provided by all reviewers. We have revised the paper accordingly and have submitted an updated version. In the appendix, we have added the following:

1. A more detailed analysis of the hyperparameters used in our data synthesis process, along with a small-scale ablation study comparing our method with greedy decoding. We also include a sensitivity analysis of the hyperparameters used in the adaptive beam size strategy.

2. An analysis of the types of problems within the MiniF2F-Test split for which the adaptive beam size strategy is effective.

3. A further examination of the limitations of existing scoring functions based on log probability, supplemented by a case study.

We would be happy to address any additional questions. If there are further concerns you would like to discuss, please feel free to contact us.

---

### Author Response · Authors · 2025-11-30
**Rebuttal Summary and Clarifications for AC Review**

Dear PCs, SACs, ACs, and Reviewers,

We sincerely thank the PCs, SACs, ACs, and Reviewers for your careful evaluation of our submission.

This note provides a concise, author-side summary of our rebuttal. Our paper targets to the research field of automated theorem proving, proposing a data-synthesis method that explores a wide range of intermediate proof states to produce diverse and informative proof steps, enabling effective one-shot fine-tuning of the LLM policy model. We also introduce an adaptive beam-size strategy that complements this synthesis pipeline by balancing exploration and exploitation during tree search of automated theorem proving.

In response to the discussion, we have added several supplementary appendices and revised parts of the paper for completeness and thoroughly address the Reviewers' concerns raised during the review process.

We believe our rebuttal thoroughly addresses the Reviewers' concerns and would lead them to form a more positve assessment of our paper. However, due to the data-leak issue on openreview.net, ACs are asked to estimate how the reviewer's impressions would have changed, incursing additional work. To assist with this process, we provide this concise summary to help the AC efficiently and accurately assess the paper and the reviewers’ feedback during the decision-making period.

For the AC’s convenience, the following parts summarize (1) the key strengths highlighted by the Reviewers and (2) how our rebuttal addresses their concerns. Detailed responses can be found in the corresponding sections of the discussion.

Thank you again for your time and consideration.

Best regards,

The Authors of Submission 2562.

---

> ### Author Response · Authors · 2025-11-30
> **Strengths**
>
> The strengths of our paper are highlighted by the reviewers:
>
> - **Novel and Technically Sound Methodology:** Introduces a **well-structured and scalable proof-state exploration pipeline** (GV4S, wuPN) that **addresses critical bottlenecks** in LLM-based ATP. Key innovations include forcing **tactic diversity** via constrained decoding and an effective decoupling strategy (yJ1K, GV4S).
>
> - **Competitive Performance and Scalability:** Achieves a **competitive 60.74% Pass@1 on MiniF2F** (wuPN) and demonstrates **practical scalability** by generating 20M samples for one-shot fine-tuning (yJ1K).
>
> - **Effective Search Heuristic:** Uses an **adaptive beam-size strategy**—a simple, interpretable linear decay—that is effective for **mitigating local traps and improving computational efficiency** without sacrificing the success rate (wuPN, yJ1K).
>
> - **High Clarity, Reproducibility, and Transparency:** The paper is **well-written and easy to follow** (oTgp). Experiments are reproducible (oTgp) with transparent computational settings (GV4S). The authors also provide **honest disclosure of limitations** and a necessary BLEU-based **decontamination step** (yJ1K, wuPN).

---

> ### Author Response · Authors · 2025-11-30
> **Summary for the response to weaknesses and questions of reviewer oTgp**
>
> The review from reviewer oTgp arrived during the rebuttal period (on 18 Nov 2025, when we had nearly completed responses for initial three reviewers.), we prepared our responses as quickly as possible to ensure a thorough and constructive discussion.
>
> - **W1: Concern regarding the data synthesis strategy.** We provide **new ablation experiments** showing that our method **consistently outperforms greedy decoding** across various beam sizes, confirming that our strategy improves data diversity and leads to a strong policy model.
>
> - **W2: Algorithmic comparison and larger budget evaluation.** We clarify that our **adaptive beam size strategy acts as a focused pruning mechanism** tailored to our policy model, and we provide futher discussion about our strategy with other approaches in tree-search methods. Furthermore, we provide **new larger budget evaluation results** for the BFS-Prover baseline on MiniF2F-Test.
>
> - **Q1: The training data used for our model and the baseline.** We clarify the understanding of the training datasets for our model and baselines (InternLM2.5-StepProver and BFS-Prover).

---

> ### Author Response · Authors · 2025-11-30
> **Summary for the response to weaknesses and questions of reviewer GV4S**
>
> - **W1: Further analysis of pruning and beam-decay hyperparameters.** We clarify that pruning hyperparameters are iteratively refine during million-scale data synthesis, and for beam-decay, we provide **new ablation results** showing that within a reasonable range, the exact choices have limited influence on performance, along with practical guidelines for selection.
>
> - **W2 part 1: Intuition behind using constrained decoding to improve data diversity.** We **supplement case studies and clarify the intuition** by demonstrating that constrained decoding is essential to prevent semantic repetition (exploitation) in search and to force coverage of low-probability core tactics (exploration), both of which are crucial for generating a high-quality, diverse, and robust training dataset.
>
> - **W2 part 2: Hyperparameter choices in the data synthesis process.** We clarify the role of each hyperparameter: alpha controls depth-based exploitation/exploration balance, beta regulates branch-wise exploitation/exploration mixture, and gamma ensures efficient resource allocation by decaying the search budget for easy problems.
>
> - **W3: Further analysis of issues in existing scoring functions.** We perform **new experiments** comparing various scoring functions, showing that performance differences between scoring functions are small, and we **provide a detailed example** illustrating that common techniques like depth normalization can sometimes hurt performance by preventing the search from abandoning unproductive subtrees.
>
> - **W4 & Q2: Generalization risks on ProofNet and handling domain shift.** We clarify that generalization problems widely exists in current Lean 4 corpus data synthesis methods, and we show that increasing the beam size partially compensates for this domain shift by increasing the chances of finding a valid proof step.
>
> - **Q1: Can the beam size be controlled by a small learned model?** We provide reasons why learning an optimal beam size controller is challenging, including its nature as a hyperparameter, the difficulty of training, and the strong model-dependent coupling of the optimal beam size.

---

> ### Author Response · Authors · 2025-11-30
> **Summary for the response to weaknesses and questions of reviewer wuPN**
>
> - **W1 & Q1: Performance of Qwen2.5-Math-7B on MiniF2F and ProofNet.** We provide **new results** showing the 7B base model's performance is currently insufficient for reliable tree-search proving, **primarily due to unreliable output formatting and a low execution success rate** of generated proof steps.
>
> - **Q2: Samples filtered during the decontamination process.** We quantify the filtering, stating that the BLEU score decontamination process **removed approximately 90k samples (0.02% of the dataset)**, and we justify the chosen threshold of 0.6 based on manual inspection of similarity distribution.
>
> - **Q3: Compatibility of the adaptive beam size strategy with other methods during the search process.** We confirm that the adaptive beam size strategy is **fully compatible with parallel expansion and batched inference** because the beam size is determined prior to the model's inference process.

---

> ### Author Response · Authors · 2025-11-30
> **Summary for the response to weaknesses and questions of reviewer yJ1K**
>
> For Reviewer yJ1K’s 13 listed weaknesses, we provide careful and detailed clarifications in the rebuttal, addressing each concern individually. While several points (such as W8, W9, and W10) do not reflect actual weaknesses of our method, we nonetheless offer thorough explanations to prevent potential misunderstandings and to better convey the intent behind our approach.
>
> - **W1 - W4: Expression issues in the abstract, introduction, figure 1, and algorithm 1.** We have **revised the phrasing and presentation** in the Abstract, Introduction, Figure 1, and Algorithm 1, and we **add an Appendix section** explaining commonly used Lean 4 tactics to enhance overall clarity.
>
> - **W5: Comparison with whole-proof generation methods.** We provide performance data for DeepSeek-Prover-V1.5 and offer a **conceptual distinction** between the two paradigms, arguing that the single-shot nature of whole-proof generation is difficult for smaller models and that its inference-time advantage is often negated when comparing comparable high sample budgets.
>
> - **W6 & W13: Contributions of the data synthesis method and the adaptive beam size strategy.** We reiterate that promoting tactic diversity and avoiding semantic repetition is critical during data synthesis by **two case studies**, and we confirm the **adaptive beam size strategy consistently improves performance** as a general-purpose, search-time optimization.
>
> - **W7: Applicability of our method to different problem types.** We confirm that **neither data synthesis nor adaptive beam size shows an inherent bias toward specific problem types**, and we present **new fine-grained evaluation results** on MiniF2F subcategories, demonstrating consistent performance improvements across all subcategories.
>
> - **W8 & W10: Further analysis of existing scoring function.** We clarify the issues in existing scoring functions and supplement **further additional experiments and a concrete case study**. In addition, we clarify that our adaptive beam size strategy can serve as a practical regulator that helps mitigate the limitations of scoring functions that rely primarily on log-probability.
>
> - **W9: Issues with different exploration strategies in tree search methods.** We clarify that our discussion is a **general observation for the ATP field**, highlighting that the **exploration component is an underexplored and valuable direction** for future community research.
>
> - **W11: About the 60 commonly used tactics set.** We clarify that the 60 tactics are selected via a top-p filtering (p=0.999) on empirical usage frequency in the STP dataset, providing a **comprehensive and practical foundation** that covers the vast majority of Lean 4 proof requirements.
>
> - **W12: Ablation study on the pruning parameters.** We clarify that the reported pruning hyperparameters are the **result of iterative refinement during million-scale generation**. While full ablations are infeasible, we explain the roles of the parameters governing depth-based exploration ($\alpha$), branch-wise mixture ($\beta$), and budget decay ($\gamma$), and how they support efficient data collection.

---

### Meta-Review · Area_Chair_tRKp · 2026-01-10

**Summary:**

-	This paper introduces a data synthesis method by providing an intermediate proof state to facilitate the one-shot fine-tuning of the LLM policy model on automated theorem proving. Their experiments show the effectiveness of the synthetic data on improving the automated theorem proving performance.

-	The main concerns of reviewers focus on the experimental part: 1) the comparison with standard search techniques (e.g., greedy search) and tree-search methods; 2) the concrete explanation on the design of constrained decoding and scoring functions; 3) the detailed data about zero-shot improvements of qwen2.5-math-7b; 4) the lack of ablation studies on parameters, such as pruning parameters.

-	During the rebuttal period, the author solved the above problems with experimental support.

-	After carefully reading about the paper itself. I found there are plenty of technique descriptions in the main content. (Introduction, methods on the proposed proof state exploration method). This content does not provide enough insight for readers to understand the meaning of these settings. Particularly without strong reference support, while it seems like an intuitive setting. Considering this is the revised version from the authors, they did not incorporate this rigorous support into the main content, whereas it is included in the appendix. Therefore, the paper lacks rigorous support in the main body, which hinders its ability to provide sufficient insight and concrete details to further the understanding of this paper.

Based on the above analysis, I reject this paper.

**Reviewer Concerns:**

All reviewers’ concerns are solved in the rebuttal section, while they are not addressed and implied in the paper revision.

**Reviewer Scores:**

•  Reviewer oTgp: No changes

•  Reviewer GV4S: No changes

•  Reviewer wuPN: No changes

•  Reviewer yJ1K: No changes

---

### Decision · Program_Chairs · 2026-01-26

Reject